# The Diffusion Model of Intra-Golgi Transport Has Limited Power

**DOI:** 10.3390/ijms24021375

**Published:** 2023-01-10

**Authors:** Galina V. Beznoussenko, Andrei Iu. Bejan, Seetharaman Parashuraman, Alberto Luini, Hee-Seok Kweon, Alexander A. Mironov

**Affiliations:** 1IFOM ETS, The AIRC Institute of Molecular Oncology, Via Adamello 16, 20139 Milan, Italy; 2School of Mathematical and Computer Sciences, Heriot-Watts University, Edinburgh EH14 4AS, UK; 3Institute of Endocrinology and Experimental Oncology, Via P. Castellino, 111, 80131 Napoli, Italy; 4Center for Research Equipment, Korea Basic Science Institute, 169-148, Gwahak-ro, Yuseong-gu, Daejeon 34133, Republic of Korea

**Keywords:** Golgi complex, Golgi apparatus, COPI vesicle, kiss-and-run, intracellular transport, diffusion model

## Abstract

The Golgi complex (GC) is the main station along the cell biosecretory pathway. Until now, mechanisms of intra-Golgi transport (IGT) have remained unclear. Herein, we confirm that the goodness-of-fit of the regression lines describing the exit of a cargo from the Golgi zone (GZ) corresponds to an exponential decay. When the GC was empty before the re-initiation of the intra-Golgi transport, this parameter of the curves describing the kinetics of different cargoes (which are deleted in Golgi vesicles) with different diffusional mobilities within the GZ as well as their exit from the GZ was maximal for the piecewise nonlinear regression, wherein the first segment was horizontal, while the second segment was similar to the exponential decay. The kinetic curve describing cargo exit from the GC per se resembled a linear decay. The Monte-Carlo simulation revealed that such curves reflect the role of microtubule growth in cells with a central GC or the random hovering of ministacks in cells lacking a microtubule. The synchronization of cargo exit from the GC already filled with a cargo using the wave synchronization protocol did not reveal the equilibration of cargo within a Golgi stack, which would be expected from the diffusion model (DM) of IGT. Moreover, not all cisternae are connected to each other in mini-stacks that are transporting membrane proteins. Finally, the kinetics of post-Golgi carriers and the important role of SNAREs for IGT at different level of IGT also argue against the DM of IGT.

## 1. Introduction

Proteins and lipids synthesized in the endoplasmic reticulum (ER) need to be transported to their sites of function. The Golgi complex (GC) has an extremely complicated structure, membrane continuity being one of the most important aspects of Golgi morphology [1,2].

Recently, we analyzed in detail the history of the development of hypotheses describing the mechanisms of transport through the GC [3], for which there is currently no consensus. Four main models of intra-Golgi transport (IGT) compete: (1) the vesicular model; (2) the compartment (cisterna) maturation–progression model; (3) the diffusion model (DM); and (4) the kiss-and-run model (KARM), which exists as symmetric and asymmetric variants [1,3]. Recently, we started a direct comparison of the models of IGT under defined experimental conditions and already compared the KARM and the cisterna maturation–progression models [2].

The DM was proposed in 1962 [4]. In 1989, Pagano et al. [5] showed that all membranes in a cell are connected and that, at any given moment, there will always be at least one continuous membrane bridge between the ER (and the nuclear envelope) and the plasma membrane (PM) passing through the GC. In 2004, we showed that, when synchronous cargo transport is restored through the GC, tubular connections are formed between Golgi cisternae [6]. The connections were found in endocytosis stimulation and signaling [7].

In 2008, Patterson et al. [8] reported that, when the GC is filled with a cargo visible under a fluorescent microscope or the same but an invisible cargo, the exit of this cargo from the Golgi zone (GZ) is described as a mono-exponential curve (exponential decay). However, a statistical analysis of the correspondence of the obtained regression lines to the theoretical regression curves was not complete and should be improved. Furthermore, the situation when transport through GC was resumed after GC was completely without cargo was not checked. In 2014, we showed that cargo proteins are observed in the connections between Golgi cisternae, while the procollagen was absent there [9]. At the same time, the speed of albumin movement from the cis-side of the stack to the trans-side was significantly higher than that of the temperature-sensitive (ts045) G-glycoprotein of vesicular stomatitis virus (tsVSVG) and procollagen (PC). At the same time, albumin was significantly enriched (an increase in its concentration per unit volume of cisternal contents). At that time, we believed that different cargo could use different models of transport. However, when inhibiting the function of coatomer I (COPI), the diffusion of tsVSVG was almost comparable to that of albumin. Moreover, not only glycosylation enzymes located in Golgi cisternae [10,11], but also proteins transferring monosaccharides into the lumen of Golgi cisternae had a lower concentration in vesicles situated around the GC than inside the cisternae per se [11]. In addition, the synchronous movement of a large amount of cargo through the GC itself changes the structure of this GC [2].

These observations suggested that there are at least a few membrane continuities between the ER, GC and the PM. These also suggest that direct membrane continuity plays a significant role in the post-Golgi transport. The DM claims that the GC represents a single membrane compartment (at least during the synchronous movement of a cargo) and cargo moves along it by simple diffusion. In order to be relevant, the DM should be based on the structures that are interconnected. The observations that do not support the DM are the following: (1) the concentrating of diffusible cargoes; (2) at each transport step, the SNARE proteins are important; (3) the rarity of connections; (4) the presence of stacks without connections during transport [6]; and (5) the deviation from a negative exponential regression line during the evacuation of cargo from the Golgi zone. Thus, the DM has significant difficulties in explaining the increasing concentrations of cargo proteins (including mega-cargoes) during EGT and IGT (see below). Furthermore, the DM cannot explain the necessity for SNAREs.

There are several observations favoring the DM, namely, in transporting Golgi stacks and after the endocytosis stimulation [7], intercisternal connections are formed [6]. When ldl F cells were heated for 2 min at 40 °C and then placed at 32 °C for 5 min, Golgi stacks are preserved and the number of intercisternal connections increased [11]. Tubular connections between Golgi cisternae have been observed even after quick freezing [6,7,12,13]. Griffiths et al. [14] described the bending of Golgi cisternae. Intercisternal connections are formed when a cargo arrives at the Golgi because Ca^2+^ is liberated from the Golgi compartments and the ER [15,16], leading to the fusion of COPI vesicles enriched in Qb SNAREs with Golgi cisternae and the restoration of the Golgi SNARE complex. These connections between Golgi cisternae are more abundant in transporting Golgi stacks and after the stimulation of cell signaling [6,7,12,17]. These intercisternal connections are permeable to small soluble cargoes, such as albumin [9], lipids [6,18], and tsVSVG tagged with GFP (VFP) [9]. However, they are impermeable for aggregates of procollagen I [11]). Moreover, dicumarol destabilizes Golgi tubules and delays IGT [19]), whereas, after the activation of protein kinase A, when the cisternae of the Golgi complex become interconnected, IGT is accelerated [20]. This suggests an important role of these connections. Some lipids can be easily transported along the secretory pathway when the formation of vesicles is inhibited [5,10,18]. When a large amount of VSVG was accumulated before the Golgi and then the transport block was released, VSVG quickly fills all medial Golgi cisternae [8,21,22]. During IGT, the lengths of the Golgi cisternae remain equal, independent of the position of a cargo within the Golgi [6,22]. Moreover, the unification of the lengths of the Golgi cisternae after the arrival of large amounts of membrane cargoes was interpreted as evidence that lipids can freely diffuse along the entire Golgi stack [6].

Patterson et al. [8] reported that a cargo that exits the Golgi area exhibits exponential kinetics. The curve-fitting of the exit of cargoes from the Golgi zone indicates negative exponentials, which suggests that each cargo molecule has an almost equal chance to exit the Golgi zone. Such types of kinetics indicate that all compartments within the GC are interconnected. However, Patterson et al. [8] did not perform careful statistical analysis to demonstrate that these curves were really negatively exponential (or correspond to the exponential decay). Furthermore, they also did not perform experiments using an empty GC. In living cells, spots filled with fluorescent cargoes can move through the pre-bleached Golgi ribbon, gradually losing their intensity [23].

Herein, we assessed the power of DM to explain the experimental data. The main question we asked in the present study was whether the DM could be the main mechanism of transport through the GC (herein, we do not examine the maturation mechanism), at least during the synchronous movement of a cargo. We applied different types of chimeric cargoes with fluorescent tags, namely, PC tagged with GFP (PFP), which can form intra-cisternal aggregates; VFP as a membrane protein that can diffuse along membranes; and albumin tagged with GFP from its N-terminus (AFP) as soluble proteins that do not form aggregates and can diffuse freely within the lumen of the Golgi cisternae and ribbon. We have checked the data obtained by Patterson et al. [8], conducting a statistical assessment of experiments similar to those and checked the situation that was not investigated by the authors, namely, the movement of a cargo through the previously emptied GC. In particular using different synchronization protocols (Table 1), we tested whether the predictions that arise from the DM correspond to the experimental data and examine whether the DM could explain the specifically designed experimental observations. We proved that, when the GC is empty, the cargo exit from GZ fits to the piecewise nonlinear curve.

## 2. Results

### 2.1. Selection of the Experimental Design

Initially, we tested whether predictions derived from the DM are fulfilled under specifically designed experimental conditions. To this end, we selected three conventional cargo proteins tagged with fluorescent parts that we presumed differ in their ability to diffuse through the Golgi compartments, namely, VFP, a conventional membrane protein [23]. PFP forms aggregates in cisternal distensions [8,24]. AFP is a soluble and highly diffusible secretory protein [9]. In general, these three proteins behaved like their natural counterparts. These GFP-tagged cargo proteins were characterized previously, and their behavior is similar to that of their untagged versions [8,9,22,23]. PFP was already used for the analysis of IGT in living cells [8]. The behavior of AFP and VFP was characterized previously [8,9]. Herein, we did not use the RUSH-cargoes [22] since these proteins irreversibly interact with biotin, not allowing the obtention of a strictly defined amount of cargo moving through the GC, and it is impossible to control the amount of cargo moving through the GC [2]. Data based on the RUSH were submitted (our unpublished observations).

Next, using different synchronization protocols (Table 1, Appendix A; see also Methods), we characterized our cargoes. Initially, we checked once more the characteristics of IGT of PFP. To this end, human fibroblasts were transfected with PFP (Figure 1A–K). Then, correlative light–electron microscopy (CLEM) was applied. At the immunofluorescence level, GFP formed dots within the GZ. These dots represented PFP-containing distensions. PFP was present exclusively inside cisternal distensions and behaved like its natural counterpart (Figure 1A–J: each consecutive image demonstrates the elements inside the box of the previous image). Consecutive frames of immunofluorescence in Figure 1J demonstrate the exit of PFP from the GZ.

### 2.2. Depletion of Cargos in Golgi Vesicles

Then, we tested whether our cargos are depleted in Golgi vesicles. To this end, we examined human fibroblasts transfected with PFP (Figure 1A–K) and HeLa cells transfected with VFP (Figure 1L,O) or with AFP (Figure 1M,P). PFP was depleted in round profiles (RPs, presumably COPI-dependent vesicles; Figure 1K: white arrows) and in COPI-coated buds (Figure 1K: black arrow). Due to the clear incompatibility of the sizes of the PFP and COPI vesicles, we did not quantify this observation because this was already established for natural procollagen I [25]. VFP (Figure 1L,O) and AFP (Figure 1M,P) were depleted in Golgi vesicles (white arrows in Figure 1L,M,O,P) but present in intercisternal connections (black arrows in Figure 1L,M). The IEM data were quantified in Figure 1N for VFP and in Figure 1Q for AFP. VFP and AFP were significantly (*p* < 0.05) depleted in Golgi vesicles (white arrows) but present in intercisternal connections. This phenomenon was demonstrated using Tokuyasu cryosections (Figure 1L,M) and enhanced nano-gold (Figure 1O,P). The concentrations of VFP (Figure 1N) and AFP (Figure 1Q) in Golgi round profiles was significantly (*p* < 0.05) lower than in Golgi cisternae in both cis- (cis-RPs and the first medial cisterna) and trans-levels (trans-RPs and last medial cisterna) of the GC. The concentration of VFP in the round profiles near the GC was lower than in Golgi cisternae independently of antibodies (cytosolic epitope or luminal epitope) and the method of IEM used (E-nano-gold or cryosections).

Thus, our cargo were not concentrated in Golgi vesicles.

### 2.3. Diffusion Mobility of the Cargos Examined

Next, we assessed the diffusion mobility of our cargo along the Golgi mass. To this end, NRK cells (Figure 1R), HeLa cells (Figure 1T,U) and human fibroblasts (Figure 1V) were transfected with AFP (Figure 1R; see also Appendix A), VFP (Figure 1T,U), and PFP (Figure 1V; Appendix A). Then part of the Golgi mass was bleached within the indicated box with the laser working at maximal intensity; images were acquired, and the ratios between the bleached areas and the areas out of the bleached boxes but within the Golgi mass were determined. Data were plotted, and the SD was estimated (Figure 1S).

The diffusion mobility of AFP (Figure 1R,S: green and red curves; see also Appendix A) is the highest, whereas the diffusion mobility of PFP was the lowest (Figure 1V,S: black line). The diffusion mobility of VFP depends on the synchronization protocol used (Figure 1S: blue and magenta lines; see [2]). When the amount of VFP transported was large, its FRAP was higher (Figure 1S,U: compare blue curve with magenta one; see also [22]) than that observed when the amount of the transported protein was small (FRAP was slower; Figure 1S,W: magenta curve). This phenomenon has been examined in a separate paper [22]. Thus, our cargos exhibited different diffusion mobilities.

### 2.4. Analysis of Models of Intra-Golgi Transport

Before the statistical evaluation of the regression lines, we performed a mental analysis of different models of IGT. The DM poses that, during IGT, Golgi cisternae are interconnected and each molecule of a cargo has almost equal possibility to leave the GC. Furthermore, no concentration of a cargo in the post-Golgi carriers would be observed [9]. Moreover when the cargo is synchronized according to the iFRAP protocol, the bleaching of the ER does not block the delivery of new portions of cargo molecules but simply makes them invisible. The invisible cargo molecules continue to arrive. Moreover, each post-Golgi carrier has equal volume and surface area (our unpublished observations; submitted) would eliminate the defined portions of cargo molecules, and no concentration of a cargo in the post-Golgi carriers would be observed.

The steady-state situation when the invisible cargo arrives at the GC, where the visible cargo departs, and equilibration of the cargo are shown in Figure 2A. The GC is considered a united membrane system wherein all cargoes diffuse freely and equilibrate their concentration along the Golgi cisternae and the stack per se. After the departure of the first post-Golgi carrier ferrying the cargo, the simultaneous arrival of a new pre-Golgi carrier and the subsequent equilibration of the cargo concentration within all Golgi cisternae would occur. Under these conditions, the number of cargo molecules eliminated by one carrier would decrease (Figure 2B).

If the GC contained a rather stable part, no concentration of a cargo in the post-Golgi carriers would occur (Figure 2C) and, after the departure of post-Golgi carriers, would decrease the Golgi size decreases and became stable (Figure 2D). According to the cisterna maturation model [1,2], the visible cargo (blue structure) arrives and then is transported along the Golgi stack without equilibration (Figure 2E). Distal Golgi cisternae are gradually transformed into post-Golgi carriers and membranes with invisible cargo are arrived at the cis- side of the GC (Figure 2F). Figure 2G illustrates the situation when the departure of GPCs depends on the growth of microtubules. Figure 2H shows a scheme illustrating the hypothesis suggesting the role of microtubule growth in the departure of post-Golgi carriers and that the fluorescence considered as the GC in reality contains several components, the GC per se and the trans-Golgi network (TGN) and immature GPCs due to the delay of the cargo exit from the post-Golgi compartment. The magenta arrow hits the DTC arc, where GPC is present. This induces the removal of the GPC present there from the post-Golgi zone. The green arrow hits the APB arc where GPC is not present. The microtubules growing from centriole (two red cylinders) could hit GPC (yellow structures; red arrow), inducing its departure or passing away (Figure 2I: green arrow; Appendix A). For the sake of simplicity, the immature GPCs are shown as a ring of dots (Figure 2I: yellow dots).

In the cells where their GCs are fragmented, the situation is as follows (Appendix A). Let us assume that, initially, we had N ministacks. Each hit and subsequent carrier departure leads to a decrease in the number of fragments and the time required for a new hit increases exponentially. Furthermore, the integrated fluorescence of the remaining Golgi fragments when the immunofluorescence of cargo in the ER is eliminated after bleaching would give an exponential decay. In cells with the fragmented GC the possible temperature or cytoskeleton-dependent reciprocating movements are short term, and their localization is stochastic (Figure 2J). Sometimes these cytoplasmic movements appear in the zone where ministacks are absent, and sometimes these movements hit ministacks leading to its movement towards the PM. This would induce the situation when the post-Golgi compartment forms contact with the PM (Figure 2J: red [hit] and green [aside] arrows). These contacts would force SNARE-dependent fusion between the post-Golgi and the PM and the disappearance of the GPC fluorescence visible in living cells. For the sake of simplicity, the cells devoid of microtubules are presented as a sphere with forces moving from outside (Figure 2K). The situation becomes inverted in comparison with that shown in Figure 2J. Under these conditions, the regression line would be described by the following formula: y = a(1 − r)x, where “a” is a coefficient depending on the size of GPCs and the frequency and intensity of the hovering of the PM and GPCs.

Using Monte-Carlo computer simulation, we demonstrated that the emptying of the Golgi area according to the exponential decay might depend on the growth of microtubules. Because, under normal conditions, the post-Golgi carriers accumulate in the post-Golgi zone, which is visible as the whole with the Golgi zone per se [2]. The average regression lines obtained after several computer trials using different parameters of the main model were similar to the exponential decay (Figure 2L–O). The curves were similar, although the slopes of their decay are different (Figure 2O). The decline of the curve obtained under different conditions (size of GPC, size of MT, frequency of MT growth) was different. However, the shape of the curve was always negatively exponential (exponential decay; Figure 2O). Thus, our analysis revealed that, in all cases, the average regression lines were similar to the exponential decay (Figure 2L–P). The negatively exponential curve of PFP exit from GZ at steady-state and during its transport through the empty GC can be explained on the basis of this mathematical model, wherein the growth of microtubules induces the exit of the carriers from GZ.

Thus, our analysis revealed that, in all cases, the average regression lines were similar to the exponential decay (Figure 2L–P). The negatively exponential curve of PFP exit from GZ at steady-state and during its transport through the empty GC can be explained on the basis of this mathematical model, wherein the growth of microtubules induces the exit of the carriers from the GZ.

### 2.5. Statistical Analysis of Regression Lines

It was difficult to find a statistically significant differences between the curves when a comparison would be made using the regression lines without their alignment at the break point corresponding to the end of the bleaching period and the beginning of the synchronization period of the cargo exit from GZ, as well as without correcting for different levels of fluorophore fading. Additionally, the level of transfection and the time it takes to prepare different cells for bleaching are variable. Thus, the heterogeneity was enormous. Therefore, when analyzing regression lines, we took into account the different degrees of bleaching and the different periods that elapsed before the proteins start to leave GZ. We added to the intensity lines the loss of intensity associated with the fading of the fluorophore and aligned the lines, taking as a common point for all regression lines the points where the line starts to bend. Taking all this into consideration, we designed statistical evaluations of the regression lines describing the exit of cargoes from GZ and assessed which of the selected theoretical curves had the highest goodness-to-fit (Figure 3A–K; see Methods).

Figure 3A demonstrates examples of regression lines. The first part of this complex regression lines fits better to the linear decline (Figure 3B). The second part of the common regression line, which we obtained after the subtraction of its first part fit better to the exponential decay-like curve (Figure 3C). The overlapping of the first and the second parts of the regression lines is shown in Figure 3D. The intersection (arrow in Figure 3D) between the linear decay (the initial part of curves and exponential decay (the second part of the curve)) indicates the point where the slope of the curve changes (the point of bending; arrow). The regression line describes the piFRAP synchronization protocol and contains the initial part similar to the linear decay followed by the exponential decay (Figure 3E). The orange line represents the exponential decay with maximal fitness for the whole regression line. The light blue line demonstrates polynomial line. The first part of the line is approximated with the liner decay (green line in Figure 3E), whereas the second part is correlated with its own exponential decay (yellow line in Figure 3E). In order to obtain the average regression line, six single regression lines were placed in one graph (Figure 3F). The variability of the experimental conditions and the ability of an operator to start the acquisition of images led to the heterogeneity of regression lines with the horizontal initial parts of different length. (Figure 3G). Figure 3H illustrates that, within the same experimental conditions but with different cargoes, the initial horizontal part could have different length. Therefore it was necessary to align regression lines using the point of bending as the uniform point (Figure 3I). The example of the average regression line obtained after the alignment of regression lines is shown in Figure 3J. Figure 3K demonstrates how different theoretical curves were estimated from individual experimental regression lines (see Table 2, Table 3, Table 4 and Table 5). The magenta curve indicates the combination of the horizontal line and exponential decline. Red lines indicate the linear decline. Green lines indicate the exponential decline. Figure 2Q demonstrates different types of exponential decay obtained after the Monte-Carlo simulation of the situation based on the fragmented GC.

As a result, we selected the following theoretical curves: linear decline, negative exponential curve, horizontal straight line turning into a linear decline, horizontal straight line turning into a negative exponential curve, and rectilinear decline turning into a negative exponential curve. Using GraphPad Prizm software (Version 9.4.0), we estimated the R_2_ and Chi2, comparing each regression line with each of three above mentioned theoretically regression lines (Table 2, Table 3, Table 4 and Table 5). Furthermore, we described details how the regression lines were corrected in the Methods.

**Figure 3 ijms-24-01375-f003:**
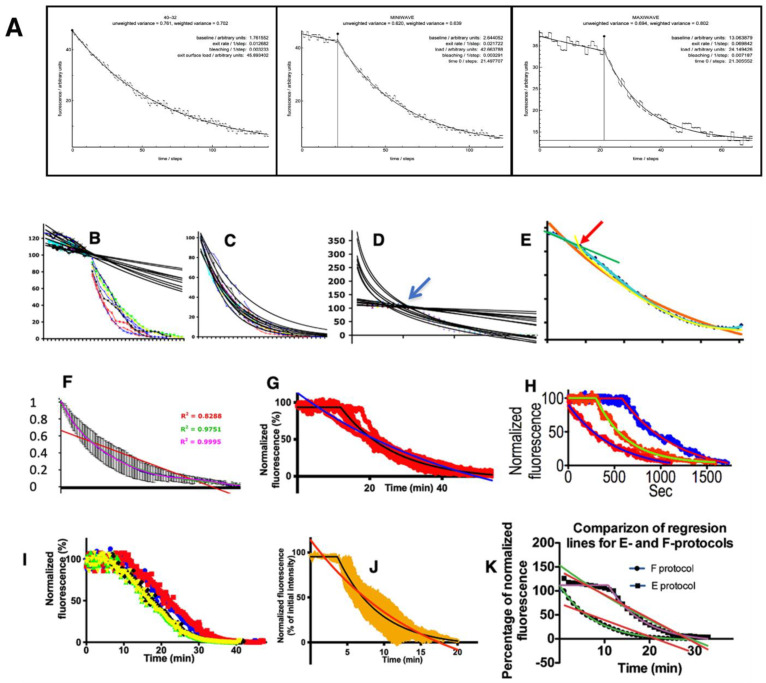
Analysis of regression lines describing the exit of cargoes from the Golgi zone. (**A**) Examples of regression lines. (**B**) The first part of this complex regression lines fits better to the linear decline. (**C**) The exponential-decay regression lines for the second parts of the curves presented. The second part of the regression line fit better to the exponential decay-like curve. (**D**) Overlapping of the first and the second parts of the regression lines. The intersection (arrow) between the linear decay (the initial part of curves and exponential decay (the second part of the curve) indicates the point where the slope of the curve changes (the point of bending; arrow). (**E**) The regression line describing the piFRAP synchronization protocol and containing the initial part similar to the linear decay followed by the exponential decay. Orange line represents the exponential decay with maximal fitness for the whole regression line. Light blue line demonstrates polynomial line. The first part of the line is approximated with the liner decay (green line), whereas the second part is correlated with its own exponential decay (yellow line). (**F**) Averaging of the regression lines describing the iFRAP protocol. Six single regression lines were placed in one graph. (**G**) Variability of the experimental conditions and the ability of an operator to start acquisition of images led to the heterogeneity of regression lines with the horizontal initial parts of different lengths. (**H**) Within the same experimental conditions but with different cargoes, the initial horizontal part could have different lengths. (**I**) The regression lines were aligned using the point of bending as the reference point. (**J**) The averaging was performed after the alignment of regression lines. (**K**) Different theoretical curves were estimated from individual experimental regression lines (see Table 3, Table 4 and Table 5). The magenta curve indicates the combination of the horizontal line and exponential decline. Red lines indicate the linear decline. Green lines indicate the exponential decline.

### 2.6. Repetition of the Patterson’s et al. Experiments

Figure 4A–F and Appendix A demonstrate the representative kymograms (the sequential regions of interest from the time-lapse recording are fused together to show dynamic changes as a two-dimensional representation) made on the basis of typical videos These kymograms demonstrate the kinetics of the exit from the GZ of our cargos, namely, AFP (Figure 4A: the E-CHM-15-CHM protocol [see Table 1: 4A]; Appendix A: the StSt-iFRAP protocol [see Table 1: 5A]), VFP (Figure 4B and Appendix A: the StSt iFRAP protocol [Table 1: 5A]; Appendix A: the E-mini-wave protocol [Table 1: 1A]; 4C; Appendix A: the E-maxi-wave protocol [Table 1: 2A]; F-40-32-40-piFRAP protocol [Table 1: 5B]. Synchronization protocol are explained in Table 1 and in Methods.

Initially, we checked whether data by Patterson et al. [8] could represent an ad-hoc observation and checked whether the curves presented by Patterson et al. [8] really correspond to the theoretical curves that are named exponential decay and could be described by the following equation: y = a(1 − r)x, where “a” is a coefficient depending on the internal parameters of the biological system (see above); “r” is a coefficient determining the slope of the curve. To this end, we repeated Patterson’s experiments and analyzed the regression lines according to our methodology. Cells were transfected with AFP (HeLa; Figure 4A), VFP (NRK and HeLa cells: Figure 4B,C and Appendix A) and PFP (human fibroblasts: Appendix A) and placed at 32 °C (steady state). Then the cells were subjected to bleaching according to iFRAP protocol and examined under a laser scanning confocal microscope (see Methods).

The analysis of the average regression lines (Figure 4G–I; Table 3, Table 4 and Table 5) demonstrated that the goodness-of-fit for curves presented by Patterson et al. when the iFRAP protocol at steady state was used is the highest for the theoretical curve named exponential decay (Table 4: 5C [for PFP] and Table 4: 6A [for AFP, PFP and VFP]). Thus, when the iFRAP protocol at steady state was used (under these conditions, the GC was filled with a cargo), the average regression lines fit better to the exponential decay.

In order to control the effect of temperature shift on the cargo kinetics and in order to test whether temperature shifts affect kinetics data, we performed control experiments. Five min after iFRAP, in the middle of the exponential curve (Figure 4F1–F3), we placed the cells on ice and immediately replaced the warm medium with the cold one. The confocal microscope continued video recording (Figure 4F4). In 10 min, we replaced the cold medium with the warm one and placed the cells back under the microscope at 32 °C. Figure 4F(F5,F6) show the movement of the sample to the previous position. Figure 4F(F7,F8) demonstrate that the incubation on ice stopped all transport. However, upon returning the cells to the permissive temperature under the microscope, the exponential kinetics began from the very same level within several seconds. (Figure 4F9) The regression line describing the experiment, during which the cell after initial imaging was placed on ice (red arrow), and then the cell was returned at 32 °C, and imaging was restored. The exclusion of the time when the cells were on ice from the plot resulted in kinetics close to exponential. The kinetics of the VFP exit from the Golgi area was not significantly affected by the temperature shifts.

### 2.7. Evaluation of the Average Regression Lines

Further, we extended the experimental design and included several additional synchronization protocols, including those from before the restoration of IGT; the GC does not contain cargo (Appendix A; Table 1). Again NRK and HeLa cells were transfected with AFP or VFP, whereas human fibroblasts were transfected with PFP. Figure 4G–U demonstrate average regression lines describing the kinetics of the cargo exit from the GZ. The mini-wave protocol for AFP did not give conclusive results. Average regression lines demonstrating the kinetics of the cargo exit from the GZ under the conditions when the GC was empty before the re-initiation of IGT, namely, AFP:CHM-15-CHM (Figure 4J; Appendix A); VFP:mini-wave (Figure 4K). VFP:maxi-wave (Figure 4L; Appendix A); PFP:CHM-15-CHM (Figure 4M); PFP:mini-wave (Figure 4N), show that parameters fit better to the theoretical curve composed of the first horizontal part followed by the linear decay. Average regression lines demonstrate that, when the 40-32-40 small synchronization protocol was used, the shape of the regression line depends on the time when the video recording started. When the GC was already filled with VFP (Figure 4O: VFP), the regression line fit to exponential decay, whereas when the GC was empty, this curve was composed of the first horizontal part, followed by the exponential decay (Figure 4P: VFP). When the mini-wave (Table 1: 40-15-40 small) protocol was used at a later stage of Golgi filling, the curve was pure negatively exponential (Figure 4Q: VFP). The use of the maxi-wave protocol did not always give unambiguous results, because, as we showed in our previous work [2], under conditions when a large amount of cargo accumulates before the GC and then synchronously releases from the block, cargo can reach the last cisterna of the medial GC within 2–3 min and begin to exit the GC.

Figure 4R–T describe the average regression lines after cargo synchronization according to the piFRAP protocol. For AFP (steady state-piFRAP), the regression contains the initial horizontal part and exponential decay (Figure 4R). Similarly to the iFRAP protocol, when the piFRAP protocol was applied to AFP, we observed a gradual decrease in the fluorescence of the Golgi mass. No spots (i.e., carriers) containing enriched AFP and moving from the Golgi were observed (our unpublished observations). When the 40-32-40 protocols were used, the shape of the regression line also depended on the initial state (empty [E] of not [F]) of the GC (Figure 4S,T).

The averaged regression lines describing the kinetics of the AFP and PFP exit from the GZ corresponded well to the third and fourth (linear decrease followed by the exponential decay) type of theoretical curves. For VFP, due to the large number of VFP arriving at the same time in the GZ, the analysis of regression lines did not give a clear answer as to which of the theoretical regression lines of the third or fourth type most fully corresponds to the averaged regression lines. The fourth type of theoretical curve exhibited the best goodness-of-fit to the average regression line describing the kinetics of VFP (Table 5; the VFP R^2^ for type 3 is less than R^2^ for type 4, and the Chi^2^ for the type 3 curve was more than the Chi^2^ for the curve of type 4), then the third type of the theoretical curves is closer in the goodness-of-fit to the PFP (Table 5; the PFP R^2^ for type 3 is more than R^2^ for the type 4 curve and Chi^2^ for type 3 is less than Chi^2^ for type 4).

### 2.8. Equilibration of Cargoes within the Golgi Complex and Kinetics of the Cargo Exit from the Golgi Complex Per Se

It was important to check whether a cargo was subjected to equilibration within Golgi cisternae. To this end, HeLa cells were subjected to the outgoing wave synchronization protocol. The cells transfected with VFP were incubated at 40 °C for 3 h. Next, cells were placed at 32° for 30 min, returned at 40° (transport block), and examined after 4, 6, 8, and 12 min. VFP was detected using an antibody against the luminal portion of VSVG and biochemical reaction based on the application of the secondary antibody conjugated with HRP and their subsequent incubation with DAB. (Figure 5A–D; quantified in Figure 5E). Figure 5E shows the dynamics of the emptying of the GC from VFP. By 7 min and 12 min, the number of the DAB-negative Golgi cisternae is significantly (*p* < 0.05) higher than before the outgoing wave (StSt). It is important to stress that, although the method of IEM-based HRP-DAB cannot allow judging whether its concentration was different in different cisternae. However, when Golgi cisternae do not contain DAB labeling, this indicates that the concentration of VFP there is rather low because the HRP-DAB method of immune EM has a very high level of amplification. Thus, VFP was not subjected to the equilibration process during its IGT.

Additionally, the outgoing wave for VFP examined on the basis of cryosections (Figure 5K–O; quantified in Figure 5R). Figure 5K shows that starting point when VFP is present on all cisternae; see [2]). Already after 4 min, a few Golgi cisternae at the *cis*-side of the stack did not contain VFP (Figure 5L). After 6 min, a significant portion of Golgi cisternae do not contain VFP (Figure 5M). After 8 (Figure 5N) and 12 min (Figure 5O), VFP was present only at the trans-side of the stack or in TMC labeled for TGN46 (Figure 5N,O; arrows in Figure 5O). Thus, cryosection-based IEM demonstrated once more that VFP was not equilibrated within different Golgi cisternae.

Graphs in Figure 5P,Q show the dynamics of the emptying of Golgi cisternae after the synchronization of PFP (Figure 5P) and AFP (Figure 5Q) correspondingly according to the outgoing wave protocol. After 7 min and 12 min, the number of the DAB-negative (see Figure 5F–H]) Golgi cisternae is significantly (*p* < 0.05) higher than before the outgoing wave (0’). Similarly, the number of cisternae not containing PFP-positive distensions (see [1] and Figure 5J]) is significantly (*p* < 0.05) higher than before the outgoing wave (0’). Figure 5R shows that during outgoing wave the emptying of Golgi cisternae was started form its cis-side. Initially, the GC was filled. Then the delivery of VFP was blocked and the labeling density of VFP was measured on Golgi cisternae in different times (see graph). Initially all cisternae were filled with VFP. By 6 min the medial cisternae at the cis-side of the stack became empty: their labeling density became significantly (*p* < 0.05) lower than at times 0’ and 2’.

If the GC according to the DM represents a united membrane system and any cargo molecule could diffuse in all directions, it is logical to expect the equilibration of the labeling density of the highly diffusible AFP. In order to check whether AFP could diffuse retrogradely being subjected to equilibration within the Golgi stack HeLa cells transfected with AFP and subjected to the CHM-15-CHM synchronization protocol. Figure 5F–H (quantified in 5Q) demonstrate the emptying of the GC filled with AFP at steady state. The delivery of AFP at the GC was blocked with CHM. AFP was detected using antibody against GFP and biochemical reaction based on the application of the secondary antibody conjugated with HRP and their subsequent incubation with DAB. No equilibration of AFP within the GC was detected.

Finally, we tested whether the cisternal distensions filled with PFP would be subjected to equilibration (Figure 5I,J [quantified in Figure 5P]). Human fibroblasts transfected with PFP were subjected to the ER accumulation–chase protocol. In 30 min, the GC was completely filled with PFP-positive distensions. Then the ascorbic acid was eliminated, and cells were placed at 40 °C in order to block the delivery of new portions of PFP at the Golgi complex. Before this shift, all Golgi cisternae contained PFP distensions (Figure 5H). In 15 min, the PFP distensions was observed only within TGN, and no distensions were visible within the cisternae (Figure 5J).

Furthermore, the kinetics of the cargo progression and its exit from the GC per se were examined (Figure 6). HeLa cells were transfected with VFP and subjected to the E-40-15-40-small (Figure 6A; Table 1, 1A). After release of the 15 °C temperature block, cells were incubated at 40 °C for 3′ (Figure 6B), 6′ (Figure 6C), 12′ (Figure 6D), or 16 min (Figure 6E) or were examined immediately after the first 15 °C temperature block (Figure 6A). Similar experiments was performed with the fibroblasts transfected with PFP (Figure 6F–H). Human fibroblasts were transfected with PFP, subjected to the E-40-15-40-small (mini-wave) protocol (Table 1, 1A), and examined using cryosection-based IEM 3 (Figure 6F), 6 (Figure 6G), and 12 min (Figure 6H) after the release of the transport block. Graphs in Figure 6I, J show that the exit from the GC per se is linear (red line; see also Table 6).

In the search among the theoretical curves, the best one in the goodness of-fit to the averaged regression lines obtained after the calculation of the regression lines for GC per se showed that the theoretical curve of type 5, namely, a horizontal plateau followed by a linear decrease, was the best (Table 6). Thus, the kinetics of the exit from the GC per se was linear.

### 2.9. Kinetics of Post-Golgi Carriers

Furthermore, we examined quantitative data describing the kinetics of the post-Golgi carriers. According to theoretical modeling, when the Golgi is filled with the visible VFP and if ER-Golgi carriers filled with invisible molecules of VFP continue to arrive at the GC (Figure 2A), then the fluorescence intensity of GPCs, which continue to depart from the GC, should gradually decrease in fluorescence intensity. HeLa cells were transfected with VFP and then subjected to the outgoing wave protocol (steady state at 32 °C for 16 h with the consecutive bleaching of the entire cell minus the GC; Figure 7A). After the departure of GPCs from the GZ, the fluorescence intensity of GPCs was almost the same as the intensity of the Golgi mass (Figure 7B: white arrows).

The ration of fluorescence intensity of GPCs versus background) was measured after synchronization according to the outgoing wave (blue regression line) and maxi-wave synchronization protocol (red regression line) and the consecutive bleaching of the cell minus the Golgi. In both cases, the intensity of GPCs was not changed significantly (*p* < 0.05) during our observation (Figure 7C). When different synchronization protocols, namely, the outgoing wave (Figure 7D: the GC was filled with VFP at pseudo-steady state [Table 2: item 3B], then the whole cell minus the GC was bleached); the empty GC, through which the large (maxi-wave; [Figure 7E]); or a small amount of VFP was moving (mini-wave [Figure 7I]), were used, the number of GPCs departing from the GC exhibited similar kinetics. The difference between the regression lines was the length of the initial part of the regression line before the beginning of the augmentation of the number of GPCS. The last part of these regression lines was similar to the exponential decay. Therefore, we estimated the best-fit curves regression lines for these parts of the regression lines describing the kinetics of GPCs. In all cases, these parts are more similar to exponential decay (Figure 7F–K).

In order to accumulate GPCs, we treated cells with tannic acid in order to block the fusion between the GPC and the PM. Under these conditions, all GPCs would be visible (Figure 7L). After the iFRAP protocol and treatment of cells with tannic acid to block the GPC fusion with the PM, GPCs left the GZ but did not fuse with the PM (Figure 7M). The regression lines describing the number of GPCs for the outgoing wave (Figure 7N) and the maxi-wave (Figure 7O) were estimated and plotted. The arrow indicates the beginning of the accumulation of GPCs.

When the mini-wave synchronization protocol was applied, the number of GPCs accumulated during the first 2 min was minimal (Figure 7P: upper row; quantified in (Figure 7Q): Mini 2′ [green bar]). Similarly when the maxi-wave protocol was applied, the number of GPCs accumulated during the first 2 min was minimal (Figure 7R: upper row; quantified in (Figure 7Q): Maxi 2′ [magenta bar]). In contrast, the GPC number was much higher when GPCs were accumulated in 9 min (P: low row; Q, R; S: Mini 9′ [red bar] and Maxi 9′ [blue bar]). When the mini-wave protocol was applied, the number of GPCs after 9 min was significantly (*p* < 0.05) higher than after two min. Furthermore when the maxi-wave synchronization protocol was applied, the number of PGCs after 9 min was significantly (*p* < 0.05) higher than after 2 min and also after 9 min during the mini-wave. Thus, the kinetics of GPCs and the amount of VFP inside them did not fit the predictions derived from the DM.

### 2.10. Lack of Intercisternal Connections in Some Transporting Mini-Stacks

We have shown that there are connections between cisternae of different Golgi compartments and that different stacks exist. They are filled with soluble cargo, which diffuses freely, and to a lesser extent with membrane cargoes [3,9,26,27]. The important question is whether cisternae are always connected during IGT. The rationale of these experiments was the following. According to the KARM, tubular connections between Golgi cisternae should not be constant. In order to address this issue, we synchronized the IGT of VFP through the Golgi mini-stacks in cells devoid of microtubules using the 40–32 °C synchronization protocol and applying EM tomography to serial 200 nm sections. We made three-dimensional reconstructions of six whole mini-stacks. Under these conditions, all mini-stacks are transporting [6]. However, in two mini-stacks, we did not find the connections. An example of the mini-stacks lacking inter-cisternal connections is shown in Figure 8A and in Appendix A. Thus, in transporting Golgi ministack, not all Golgi cisternae are connected.

### 2.11. Role of Membrane Fusion Coatomer I, and Stability of Intercisternal Connections for Intra-Golgi Transport

Further, we tested the role of membrane fusion and the stability of tubular connections between Golgi cisternae. The rationale of these experiments was the following. In our previous paper [2], we showed that COPI is important for the formation of pores and Golgi tubules. On the other hand, one of the most important predictions of the KARM is the transiency of the connections between Golgi cisternae. Previously, we demonstrated that albumin uses diffusion for its movement through the GC along intercisternal connections formed during IGT [6,12]. These connections are filled with soluble cargos and could be used as a pathway for the intra-Golgi transport of such cargoes [9]. In order to examine the role of intercisternal connections, we used two approaches, namely, we blocked membrane fusion and unnecessary Golgi vesiculation using the simultaneous addition of NEM and brefeldin or the microinjection of α-SNAP mutant together with the inhibitory antibody against ßCOP. The rationale of these experiments is the following. We showed previously that, when membrane fusion is blocked, the Golgi is converted into a number of 52 nm vesicles, whereas Golgi cisternae became highly invaginated [28,29,30,31].

Initially, we assessed the role of COPI for the concentration of AFP. To examine the role of functional coatomer I, control CHO cells (Figure 8B,C) and ldl F cells (Figure 8D,E) transfected with AFP were heated at 40 °C for 2 min to block the ability of COPI in ldl F cells to form vesicles [11,22]. In the heated control CHO cells, the AFP concentration of AFP within the GC increased (Figure 8C) whereas in ldl F cells, this concentration was almost invisible (Figure 8E). Thus, a functional coatomer able to form COPI-dependent vesicles is important for the concentration of AFP.

Next, the role of COPII cells transfected with AFP or VFP and synchronized according to the iFRAP protocol was examined, and the ratio between the fluorescence intensity after FRAP and the original level was plotted. (Figure 8F). Orange circles: AFP:CTR (control: no treatment with inhibitors). Blue squares: cells transfected with AFP under the action of FLI-06. Magenta triangles: treatment with NEM/BFA of the AFP transfected cells. Green triangles: the VFP transfected cells without application of the inhibitors. Black rhombs: the VFP transfected cells treated with FLI-06 (the inhibitor of COPII; see [24]). Red rings: the VFP transfected cells treated with NEM/BFA. Thus, we revealed at the level of light microscopy that the blockage of membrane fusion (NEM/BFA) significantly (*p* < 0.05) blocked the concentrating of AFP and VFP within the GC, whereas the inhibition of the COPII function with FLI-06 did not significantly affect (*p* > 0.05) this process.

Additionally, we tested the role of SNAREs at the level of electron microscopy. Treatment of cells with NEM/BFA inhibited the IGT of PFP distension (Figure 8G). This treatment had a lower effect on the IGT of AFP (Figure 8H,J) and VFP (Figure 8I).

HeLa cells were transfected with AFP. The microinjection of these cells with an irrelevant protein did not affect the IGT of AFP. The microinjection of the AFP-transfected cells with α-SNAP mutant inhibited the delivery of AFP at the trans-side of the GC after AFP synchronization, according to the CHM-15-CHM protocol (Figure 8L: compare with Figure 8K). Arrows show labeling for ERGIC53 (cis-Golgi marker).

When we microinjected the cells with the mixture of an anti-ß-COP antibody and α-SNAP mutant and in 3 min prepared cells for light microscopy in electron microscopic analysis, we obtained similar results (Appendix A). Two representative examples of experiments are shown in Appendix A. The microinjection of α-SNAP mutant into HepG2 cells blocked the entrance of albumin into the GC from the ER (Appendix A). The microinjection (green cells) of an α-SNAP mutant into HepG2 cells inhibited the exit of albumin from the ER (Appendix A). The microinjection of an irrelevant protein (anti-albumin antibody) did not block the exit of albumin (red in [Appendix A], green in [Appendix A]) from the ER, indicating that, in control cells (the irrelevant microinjection), the delivery of AFP at the GC was normal (Appendix A). Thus, in the presence of the α-SNAP mutant and anti-ßCOP antibody, we observed the slowing down of the ER–Golgi transport of AFP.

Finally, in order to understand whether the stability of tubular connections between Golgi cisternae is important for IGT within the framework of the DM, we used inhibitors of cytosolic phospholipase A2α (cPLA2α). It is known that inhibitors of PLA2 destabilize these tubules. Their application leads to the disappearance of intercisternal connections in Golgi stacks [32]. To destabilize these connections, we pre-treated cells with the inhibitors, namely, pyrrophenone, which is potent and specific for cPLA2α with an IC50 value of 4.2 nM, or ONO. ONO-7300243 is a potent lysophosphatidic acid receptor 1 antagonist with an IC50 of 0.16 μM [33].

In HeLa cells transfected with AFP, pyrrophenone (the inhibitor of cPLA2α) treatment inhibited the delivery of AFP at the trans-side of the GC (Figure 8M–O). Arrows show the labeling for ERGIC53 (cis-Golgi marker in Figure 8M]) and for GM130 (cis-Golgi marker in Figure 8O]). Additionally, we treated the AFP (Figure 8P) or VFP (Figure 8Q) synthesizing HeLa cells with ONO. The treatment of cells with ONO, another inhibitor of cPLA2α, blocked the delivery of AFP (Figure 8P) or VFP (Figure 8Q) at the *trans*-side of the GC in the correspondingly transfected cells.

Thus, the destabilization of tubular connections between Golgi cisternae blocked IGT all our cargoes along the Golgi stack. Altogether, our results do not support the DM.

## 3. Discussion

Herein, we examined the ability of the DM to explain experimental results related to IGT. For this purpose, we used different methods to synchronize cargo transport through the Golgi complex and three different types of cargoes tagged with GFP, namely, a membrane protein (VSVG), a soluble protein (albumin), and a soluble protein that forms aggregates (procollagen). We did not use the RUSH system because, after the addition of biotin, its binding to cargo is irreversible [22]. This issue is examined in a separate paper. Furthermore, we demonstrated once more (see [26]; reviewed in [1,2,3,26,27,28,29,30,31,32,33,34,35,36,37]) that these cargoes were depleted in round profiles (presumably COPI vesicles) and that COPI-coated buds contained lower concentration of anterograde cargoes than Golgi cisternae. Together with the discrepancy between the size of the vesicles and mega-cargoes, this argues against the DM.

Initially, we selected three routinely used cargoes visible in living cells and checked data obtained by Patterson et al. [8]. Initially, we proved once more than these cargoes were excluded (at least partially) from Golgi vesicles. Additionally, we demonstrated that these cargoes had different diffusion mobility. Next, we analyzed the situation mentally and found several theoretical curved suitable for our analysis.

To explain why the exit of the almost non-diffusible PFP from GZ occurred according to exponential decay, whereas its exit from the GC was linear, we proposed a hypothesis able to explain why Patterson et al. [8] demonstrated that the exponential decay (decline) is suitable for the analysis of the cargo exit from GZ when the GC is filled with a cargo. Next, we designed a mode suitable for the statistical analysis of the regression lines obtained for different cargoes and under different situations.

Since the computer simulation showed the possibility of using a negative exponential curve to describe the exit of various types of cargoes from GZ, we chose five theoretically possible curves for our analysis to compare them with the obtained regression lines. This set included a linear decrease in values, an exponential decrease in values, a curve consisting of an initial horizontal section that would be replaced by a negative exponential decrease, and a curve consisting of an initial horizontal section that would turn into a linear decrease in values.

Our analysis of the curves demonstrated that, when the GC is filled, the cargo exit curve was similar to the exponential decay, whereas, when the GC is empty, the kinetics of the cargo exit from GZ could be explained with the curve composed of the initial horizontal part followed by the exponential decay. However, when the cargo exit from GZ was examined using the outgoing wave protocol, we did not observe the equilibration of our cargoes within the Golgi cisternae in stacks. Additional analysis of the kinetics of the post-Golgi carriers revealed that a cargo did not equilibrate within the Golgi stack. Moreover, the exit of the cargoes from the GC per se fit mostly to the linear decline. All these data indicated that the DM is not suitable for their explanation.

The regression line describing the emptying of GZ from these cargos was dependent on the transport state of the GC. When the Golgi stacks and the Golgi zone were filled with any of our cargoes, the exit of the respective cargoes from the Golgi zone occurred according to a negatively exponential regression curve (exponential decay). In contrast, when the GC was empty, the kinetics of the cargo exit were composed of two parts, namely, an initial horizontal line followed by the exponential decay. Control experiments revealed that the temperature shift had minimal influence on the curves. The curve describing the exit of these cargoes from the Golgi stacks per se exhibited either linear decay when the GC was full of cargo or piecewise nonlinear regression when the first segment was horizontal, followed by the second segment being similar to exponential decay. This suggests that IGT does not depend on the rate of cargo diffusion.

Therefore, we additionally tested some predictions derived from the KARM. This allowed us to demonstrated that SNAREs are important for the ER–Golgi and IGT. Even transporting a stack in some moment did not have an intercisternal connection (Appendix A), indicating that these connections are temporal. The stability of these connection is really important, because their destabilization blocked IGT. The stability of the connections also depended on COPI.

Under normal conditions, the post-Golgi carriers are accumulated in the post-Golgi zone [2]. It is not possible to discriminate between GPCs and the GC at the level of light microscopy [3]. Therefore, when the iFRAP or piFRAP protocols were used for the evaluation of the regression lines describing the emptying of the GC, in reality, these curves describe the exit of the cargo from the Golgi zone, which includes the GC and TGN and GPCs. Using Monte-Carlo computer simulations, we demonstrated that the emptying of the Golgi area according to the exponential decay might depend on the growth of microtubules or the hovering of GPCs in mini-stack system.

Importantly, it seems that the exit of cargoes from the TGN is a rate-limiting step in the transport of cargoes (our unpublished observations to be submitted). In order to understand why there is a delay in transport at the level of the post-Golgi compartment, it should be remembered that the delivery of post-Golgi carriers from the centrally located GC to the PM is carried out using kinesin moving the carrier through the microtubule (MT) [38,39]. At the same time, microtubules are constantly destroyed and grow again from the center of the MT organization. The carrier is delivered to the periphery only if the mature carrier has received kinesin from the TGN structures where kinesin is localized [40] and if MT passes close to the carrier. The growth of MTs does not depend on the state of the GC. Therefore, if the dispatch system located at the post-Golgi compartment level is overloaded, then the growth of MTs will be a limiting factor for sending carriers to the periphery.

A similar situation arises in the case when microtubules are depolymerized. In such a situation, the post-GC carrier will fuse with the PM only if it is situated close to it, since structures with a diameter of more than 50 nm cannot diffuse in the cytosolic gel [41]. Therefore, it is necessary to reorient GPC to a position close to the PM. That is why, if you add nocodazole in order to depolymerize microtubules immediately after the start of transport through the GC, you get a long delay in the delivery of carriers to the PM. In particular, for carriers containing procollagen, the delay can reach 4 h [28]. Only after the central GC is completely fragmented and ministacks are located along the PM will the limiting factor ensure that the mono-exponential type of cargo leaving the Golgi stack will be associated with random temperature-dependence or be dependent on other factors, i.e., on the hovering of the ministacks or the PM. These approaches occur randomly and with a certain frequency and do not depend on cargo transport.

In general, there were two main discrepancies: (1) between the curves containing the exponential decay parts when the Golgi zones was examined and the curve containing linear decay parts when the GC per se was examined and (2) the emptying of the Golgi zone also being independent of the cargo diffusion mobility. This forced us to perform the Monte-Carlo computer-based simulation (modeling). To this end, we suggested that the exponential decay could be a result of microtubule growth and interaction between MT and GPC when MT hits GPC. Furthermore, we evaluated whether different sizes of and GPCs and different distances between centrosomes and the GPCs would affect the shape of the curves. After arising from the centrosome, microtubules come into contact with the carriers, thus giving them a signal or providing them with a mechanism to exit from the Golgi area. In cells with the central GC, post-Golgi carriers leave the Golgi zone only when they attach to microtubules (Appendix A).

The ability to form intercisternal connections and the stability of these connections are important for the diffusion of our cargoes. However, for the concentration of diffusible cargo, these connections should be temporal [30,31]. The rationale of these experiments is the following. We showed previously that, when membrane fusion is blocked, the Golgi is converted into a huge number of 52 nm vesicles, whereas Golgi cisternae became invaginated [29,32]. Therefore, for the immobilization (“freezing”) of the Golgi morphology, we treated cells with the mixture of NEM and brefeldin A, whereas, for the destabilization of intercisternal connections, we used ONO and pyrrophenone [32,33,34]. Previously, we showed that SNAREs are necessary for IGT and that, when a large amount of a cargo moves across the GC, this cargo reaches the last medial cisterna but never TMC [2].

The following data do not support the DM: the independence of the cargo-exit kinetics from the ability of the cargoes to diffuse along the Golgi ribbon, the kinetics of the exit of these cargoes out of the empty Golgi, the linear kinetics of cargo exit from the Golgi stacks, the concentration of albumin inside the *trans*-most cisterna, and the concentration of cargo aggregates that cannot diffuse along thin inter-cisternal connections in the trans compartments of the Golgi complex during intra-Golgi transport.

Thus, all arguments (see Introduction) in favor of the DM could be explained in other ways. The kinetics of cargo exit from the GC per se and the Golgi zone depended on whether the GC or Golgi zone are filled with a cargo. The enrichment of PCI-containing immature GPCs in fibroblasts and lipid particles at the *trans*-side of the Golgi in hepatocytes and enterocytes and the concentration of GFP-albumin at the *trans*-side of the Golgi stack argue against the DM [2,37,42,43]. At steady state, most PCI and VSVG localize at the trans-side of the Golgi and within the Golgi-to-PM carriers [2]. In organs, most PCI distensions within the Golgi area are not integrated into the Golgi cisternae but are localized externally to the stacks, apparently within the putative post-Golgi compartment or TGN [25]. Furthermore, the transient existence of transporting stacks without inter-cisternal connections at defined times argues against the DM.

The kinetics of the exit of cargo from the GZ differed from that of the kinetics of the cargo exit from the GC per se and appeared to be linear. The deep penetration of VSVG across the Golgi stacks was not observed when a small amount was used [2,3]. Membrane fusion was necessary for the progression of a cargo across the Golgi stack. However, the rapid inhibition of εCOP accelerates the cis-to-trans passage (diffusion) of VSVG [11].

The DM of IGT has several other problems: the protein, lipid, and ionic gradients across Golgi stacks; the presence of SNAREs within all steps of IGT; the secretory pathway concentration of albumin [15]; regulatory secretion cargoes [44]; and mega-cargoes not able to diffuse along narrow intercisternal connections [2]. Albumin reaches the *trans*-side of the GC faster than procollagen and VSVG [9]. Already by 2 min after the release of the transport block, the concentration of albumin in the TGN became 9-fold higher than in the ER (see Figure 1I by [9]) and reaches 9–10-fold in 3–5 min. Furthermore, the tripeptide moves along the Golgi stack rather quickly [45]. Intercisternal connections are permanent for albumin [9] and therefore (if all cisternae the are interconnected) enrichment of albumin at the trans-side of the GC should not be observed. These data argue against the DM.

In order to inhibit the formation of intercisternal connections, we treated cells with BFA and NEM simultaneously or after the simultaneous microinjection of the αSNAP mutant and anti-ßCOP antibody [29]. This blocks membrane fusion and prevents Golgi vesiculation. In order to destabilize ICs, we treated cells with PLA2 inhibitors, namely, pyrrophenone or ONO, which lead to the disappearance of ICs [32]. In all these cases, IGT became slower, and the exchange of SNAREs was blocked. There are several other observations in favor of the important role of SNARE for IGT in situ. It is known that Rabs regulate membrane fusion [46]). In *P. pastoris*, calcium/calcium-permeable ion channels are important for IGT and the stacking of the Golgi cisternae [47]. The importance of membrane fusion for IGT also argues against the DM.

Our data on the intra-Golgi transport of PFP at steady state confirmed results by Patterson et al. [8]. In order to find an explanation, we estimated the curves according to which PFP exited from the Golgi per se. Thus, the obvious hypothesis would be that, in our experiments and in experiments by Patterson et al. [8], the Golgi mass represented not only the GC per se but also some post-Golgi carriers. PCIII-GFP behaved similarly ([24]; our unpublished observations). Patterson et al. [8] observed the deep penetration of VSVG-GFP into Golgi stacks after the release of the 40 °C temperature block and the exit of PFP from the Golgi zone according to the exponential decay. The authors consider this zone to represent the GC per se. If so, this suggests that PCI-containing distensions freely diffuse along the united lumen of the Golgi cisternae. However, PFP cannot diffuse along connections between cisternae due to size limitations [1,9]. To resolve this contradiction, Patterson et al. [8] proposed that (1) this mega-cargo (PFP), which diffuses slowly, could exit from any part of the Golgi stack, including the *cis*-side, and thus, the notion of IGT, i.e., a directional transport from the *cis*- to *trans*-side of the Golgi, is an artefact of protocols used to study the intra-Golgi transport; and (2) there was no preferential exit of the cargoes based on their time of arrival, i.e., the cargoes that arrive early do not preferentially leave early. Patterson et al. [8] suggested that this indicates that, in contrast to other cargoes, PFP leaves the GC immediately after its arrival at the GC even from a *cis*-Golgi cisterna. However, Patterson et al. [8] did not examine the GC, which was empty before the restoration of IGT. Bleaching per se does not eliminate cargo from the GC. Thus, the arrival of VFP was not blocked. They simply made this cargo invisible. Furthermore, the process of the GC filling with cargos could take a significant time, and, under such conditions, the exit kinetics could be different.

If we take into consideration that PFP arrives to the *cis*-Golgi cisterna, this means that PFP left the Golgi from the *cis*-cisternae of the Golgi stacks. However, previous electron microscopic data [25] and our current observations suggest that procollagen I is transported through the cisternae of the Golgi stack in less than 20 min. Moreover, application of the cargo out-wave [27] demonstrated that PFP first leaves the *cis*-side of the Golgi complex and only then exits from the *trans*-side. Recently, we demonstrated that the deep penetration of a cargo up to the last Golgi medial cisterna is determined by its amount [2]. Our data demonstrated that the exit of VFP and PFP-containing distensions from the Golgi per se occurred according to a linear regression line. Furthermore, there is the accumulation of VFP at the trans-side of the Golgi at steady state [2,3]. Herein, we demonstrated that, when membrane fusion was blocked, there was no exit of cargo from the *cis*-side of the GC and no progression of cargoes through Golgi stacks.

The DM could be improved if the idea by Griffiths [48] that peristaltic movement might be involved in IGT would be assumed. However, the size of mega-cargoes and the absence of actin–myosin machines around intercisternal connections argue against this proposal. The simple diffusion along permanent connections between Golgi compartments cannot be an efficient mechanism for the concentration of cargoes at the trans-side of the GC. Of interest, the KARM easily explains the concentration of albumin, VSVG, mega-cargoes, and regulated secretory proteins along the *cis*-to-*trans* direction of the stacks [3,27,35,49,50,51]. However, this aspect was beyond the scope of our study.

Analysis of the graphs of the decrease in the intensity of the fluorescence of the VSVG-GFP in the area where the GC is located, shown in Figure 1I, by Patterson et al. [8]) revealed that the initial section of their graph has a practically linear shape and only then is replaced by the curve similar to the exponential decay. This type of curve corresponds more to a two-phase regression line composed of an initial linear section and its subsequent mono-exponential decrease. On the other hand, the regression line reflecting the corresponding decrease in ss-GFP fluorescence after an identical synchronization protocol almost perfectly corresponds to a decrease in the fluorescence of the mono-exponential type (Figure 1E: ss-YFP by Patterson et al. [8]). Unfortunately, the authors do not provide data on the statistical comparison of the correspondence of the presented regression lines to the types of curves indicated by us. Therefore, we cannot claim that our conclusion is more correct than the one made in the article. However, taking into account our own data, such an interpretation explains the above contradiction.

The fact is that, with the massive delivery of VSVG at the GC, the so-called cargo ribbon arises [2]. If this is the case, then VSVG can diffuse very quickly at the last medial cisterna of the GC [2,8,52,53,54]. These portions of cargo are located at the distal part of the medial GC and are sent from the GC in the initial period and in different types of cells with different intensities. This leads to a linear decrease in fluorescence in accordance with the maturation model of transport. In order to understand why Patterson et al. [8] claim that regression line describing the emptying GZ after the piFRAP synchronization protocol, we examined this issue more attentively and divided our graphs into two parts and calculated R^2^ and Chi^2^ using the theoretical curve composed of the initial linear decay and then exponential decay. In the case of the piFRAP synchronization of VSVG-GFP, R squared was maximal, and the sum of R squared was minimal only after this assumption. In contrast, linear decay, exponential decay, and the complex curve consisting of a horizontal part and then exponential decay exhibited worse values. When we applied the piFARP synchronization protocol to PCI-GFP, the resulting regression line exhibited higher fitness to the curve composed of the initial horizontal plateau followed with exponential decay (Table 5).

Summarizing herein, for the first time, the kinetics of the exit of various commonly used standard cargoes from the Golgi zone was studied under conditions when the synchronization of the cargo wave begins under conditions when the GC does not contain cargo. It is proven that the kinetics of the passage of various cargoes through the GC corresponds to a straight regression line. The observations that there is no balancing of cargo concentration between different cisternae are completely original. It is shown for the first time that transport through different CG compartments requires the SNARE function; it is established that the connections between different GC cisternae are not constant, which rejects the diffusion pure model and calls into question the universality of using a negative exponential curve for cargo exiting the GC zone. Our data argue against the DM.

## 4. Materials and Methods

### 4.1. Reagents

Unless otherwise stated, all of the chemicals and reagents were obtained from previously indicated sources [55,56] or from Sigma (Milan, Italy). FUGENE6 transfection reagent was from Roche (Monza, Italy). The following antibodies were used: polyclonal against GM130 (from M. A. De Matteis, Mario Negri Sud Institute, Italy); monoclonal against ERGIC53 (from H. P. Hauri, University of Basel, Switzerland); polyclonal against the luminal and cytosolic domains of galactosyl transferase (GalT; from E. Berger, University of Zurich, Switzerland); polyclonal against ßCOP (from J. Lippincott-Schwartz, National Institutes of Health, Bethesda, USA); polyclonal against ManII (from K. W. Moremen, University of Georgia, USA); polyclonal against the α1 chain C-terminal domain of PCI (from L. W. Fisher, National Institutes of Health, USA); monoclonal against the VSVG luminal domain (from J. Gruenberg, University of Geneva. Switzerland); polyclonal against the luminal domain of tsVSVG (from K. Simons, Max-Planck Institute, Germany); monoclonal against VSVG (P5D4; from Sigma-Aldrich, St. Louis, MI, USA); polyclonal against TGN46 (from S. Ponnambalam, Wellcome Trust Centre for Human Genetics, Headington, Oxford, UK); polyclonal against ßCOP (from Agilent/DAKO, Santa Clara, CA, USA); polyclonal against GFP (from Abcam, Cambridge, UK); and polyclonal conjugated with Alexa 488 and Alexa 546 against rabbit, mouse and sheep IgGs (from ThermoFisher Scientific, Waltham, MA, USA). Fab fragments of polyclonal antibodies against these IgGs were from Jackson ImmunoResearch (West Grove, PA, USA), nano-gold-conjugated Fab fragments of anti-rabbit IgG and Gold Enhancer were from Nanoprobes (Yaphank, NY, USA), and protein A conjugated with colloidal gold was from J. Slot (Utrecht University; The Netherlands). Pyrrophenone was from MedChemExpress (Princeton, NJ, USA; catalog no.: HY-111376).

### 4.2. Constructs

Reagents used in this study are presented in Table 7. J. Lippincott-Schwartz (Janelia Research Campus, Howard Hughes Medical Institute, Ashburn, VA) kindly provided the cDNA of GalT-GFP and VSVG-YFP. AFP [9] and the procollagen III (PCIII)-GFP and PCI-GFP (PFP) constructs [21,24] were characterized previously. PCI tagged with GFP at its C-terminus did not work in our hands, and therefore two constructs were produced: the α1 and α2-chains of PCI both tagged with GFP at their N-terminus. Briefly, the leader sequence of collagen type I α2 was excised by PCR using the forward primer CGGCTAGCATGCTCAGCTTTGTGGAT and the reverse primer CGACCGGTTGGCATGTTGCTAGGCATAA and then restriction-cloned at the 5’ end of GFP in the pEGFPC3 vector (Clontech, Mountain View, CA, USA), using the NheI and AgeI sites. This modified vector was used for the sub-cloning between the HindIII and the BamHI sites of the remaining collagen type I α2 sequence, which was excised by PCR using the forward primer GCGAAGCTTCAATCTTTACAAGAGGAAACTGTAAG and the reverse primer GCGGGATCCTTATTTGAAACAGACTGGGCCAATGT. The same modified vector was used as for the sub-cloning of collagen type I α1, between the HindIII and PstI sites, after PCR-excision using the forward primer GCGAAGCTTCAAGAGGAAGGCCAAGTCGAGG and the reverse primer GCGCTGCAGTTACAGGAAGCAGACAGGGCCAACG.

### 4.3. Cells

Human fibroblasts were from M. De Luca (Istituto Dermatopatico dell’Immacolata, Rome, Italy). HeLa cells, NRK, and HepG2 cells (from the same sources as described previously) [9,57] and human embryonic kidney 293T cells (from ATCC [293T (ATCC^®^ CRL-3216™]) were cultured in Dulbecco’s modified Eagle’s medium supplemented with 10% fetal bovine serum, 50 mg/mL penicillin, and streptomycin in a 5% CO_2_ incubator. Cells were grown in A-DMEM/F12 (1:1), 5% fetal bovine serum, 4 mM GlutaMAX (all from Gibco), 100 U/mL penicillin, and 100 μg/mL streptomycin (from Sigma-Aldrich). In each experiment, 6 pairs (control and experimental one) of dishes with cells were used, and each dish was fixed on a different day. We always used 32 °C and two conditions: (1) embryonic serum and (2) adult serum as a control.

In order to prevent Golgi vesiculation, we used brefeldin A together with NEM, as was described by [29].

### 4.4. Designs of Experiments

Transfection of human fibroblasts with PFP was performed by electroporation (1.5 µg cDNA per 2 × 10^5^ cells; V, 400 v; C, 870 µF) or as described previously [8]. Cells were transfected with GFP-albumin, VFP, and PFP exactly as was described previously [9]. Transfection of cells with VFP and GFP-albumin was carried out using electroporation, as described previously [15]. The experiments based on the siRNA interference were organized as before [57]. Transient transfections of cells were performed 16 h prior to experimental imaging. The α-SNAP mutant was injected into the cell, then the cell was transferred to ice for 10–15 min so that the antibodies diffused through the cytosol. Then the cells were transferred to a medium at a temperature of 37 °C, and 5 min after heating, they were prepared for analysis using immunofluorescence.

The infection of the cells with the 045 temperature-sensitive strain of VSV and their transfection with the different fusion proteins (only cells with low levels of transfection were examined) and the stimulation of PC and PFP synthesis in human fibroblasts were all performed as described previously [9]. The synchronization of intra-Golgi transport was performed using different protocols derived from previous studies [3,6,17,26]. The synchronization protocols used were without or with cycloheximide (CHM; 150 µg/mL; to block protein synthesis in the ER) and are summarized below, with further details given in Table 1 and Appendix A. Briefly:

1. The pulse (40-15-40) protocols could be small (the mini-wave; Table 1, 1A,B) or large (the maxi-wave; Table 1, 2A,B). In both cases, the cells were transfected for VFP (HeLa cells) or PFP (human fibroblasts) (AFP is not suitable for these protocols) and kept at 40 °C for 16 h in the absence of ascorbic acid to accumulate VFP or PFP in the ER. Then, the cells were shifted to 15 °C for 15 min (Table 1, 1A,B) or 2 h (Table 1, 2A, B) and then were shifted back to 40 °C in the confocal microscope stage, when the video recording was started (sometimes the whole cell minus its Golgi complex was also bleached).

2. The pulse (40-32-40) protocol (Table 1, 3A,B). HeLa cells were transfected for VFP and placed at 40 °C for 16 h, then shifted to 32 °C for 5 min, and then back to 40 °C in the confocal microscope stage. Finally, the video recording was started (sometimes the whole cell minus its Golgi complex was also bleached). The rationale of these protocols is to allow a defined (small or large) amount of a cargo to move through the GC.

3. The emptying pulse [32(CHM)-15-32(CHM)] protocols (Table 1, 4A,B). The cells were transfected for one of the cargoes and placed at 32 °C for 16 h. Then the cells were treated with CHM for 3 h at 32 °C to empty the Golgi complex and the ER. The cells were then placed at 15 °C for 2 h, and the CHM was removed to initiate protein synthesis and the transport of the cargo to the pre-Golgi compartment. Finally, the cells were placed at 32 °C in the presence of CHM when the video recording was started (sometimes the whole cell minus its Golgi complex was also bleached). The rationale of this protocol is allowing a defined amount of any cargo to move through the GC.

4. The pulse (40-32-40) protocol (Table 1). HeLa cells were transfected for VFP or PFP and placed at 40 °C for 16 h to accumulate VFP or PFP (also in the absence of ascorbic acid) in the ER. Then the cells were shifted to 32 °C for 5 min, and next, the cells were placed at 40 °C again. The rationale of this protocol is to allow a small amount of cargo (mostly tsVSVG and PC) to move through the GC.

5. The iFRAP-steady-state protocol (Table 1, 5A). The cells were transfected for one of the cargoes and placed at 32 °C for 16 h. Then the whole cell minus its Golgi complex area was bleached, and the video recording was started. This is a repetition of Patterson et al.’s protocol. It would allow us to test the fitness to exponential decay statistically.

6. The piFRAP-steady-state protocol (Table 1, 5B). The cells were transfected for one of the cargoes and placed at 32 °C for 16 h. Initially, the Golgi complex was bleached, and after 3 min, the whole cell minus its Golgi complex area was bleached again, and the video recording was started. This is a repetition of the Patterson et al. protocol. It would allow us to test the fitness to exponential decay statistically and compare this fitness with the regression line composed of a linear decay and an exponential decay.

7. The outgoing wave protocol (Table 1, 6A). The cells were transfected for VFP or PFP and placed at 32 °C for 16 h. Then the cells were placed at 40 °C in the confocal microscope stage, and the video recording was started [6,9,22]. This protocol would allow us to evaluate whether cargo would equilibrate its concentration within the Golgi stack.

### 4.5. Light Microscopy

The fluorescence microscopy analysis included time-lapse analysis, with the measurement of fluorescence recovery after photo-bleaching (FRAP) carried out using a confocal system (LSM510 META; Carl Zeiss, Oberkochen, Germany). The cells were fixed at the desired times using 4% paraformaldehyde and then processed for immunofluorescence. Cells on coverslips were analyzed with this confocal microscope with an objective 40× lens using the original manufacturer’s software. The GFP fluorescence was bleached by 50 pulses for a total time of 0.07 s at 90% laser power/100% transmission. The fluorescence intensity (*F*) in the bleached area was monitored as a function of time after the bleaching by scanning at 90% laser power/0.5% transmission every 3 s. In our hands, under these conditions, the photo-bleaching was irreversible. The level of the co-localization of two markers was measured as described previously [9,39].

For FRAP analysis, HeLa cells were transfected with VFP and AFP, whereas human fibroblasts were transfected with PFP. Then, in the transfected cell, part of the Golgi mass was bleached within the indicated boxes indicated in Figures with the laser working at maximal intensity; images were acquired, and the ratios between the bleached areas and the areas out of the bleached boxes but within the Golgi mass were determined. Data were plotted, and the SD was estimated.

All live-cell imaging was performed on cells grown in MatTek chambers and maintained in a CO_2_-independent medium. The brightness and contrast of the fluorescent images were adjusted (Photoshop CS5.1), and all of the live confocal images and the wide-field images were Gaussian-filtered (rotationally symmetric low-pass filter). Only dots with intensity at least double the background were counted. For immunofluorescence and immune EM, we performed all of the necessary controls, including the control of the secondary antibodies and the positive and negative controls. When control cells are mentioned, these were the cells that were the required control of the experiment cells. These will have included irrelevant microinjection, control RNA oligonucleotides, and other such relevant controls.

The various cell treatments included 2.5 µM FLI-06, 0.3 and 5 µg/mL BFA, [2], and 1 mM NEM, as described in [2,29].

### 4.6. Electron Microscopy

For electron microscopy (EM), the samples were prepared as described previously [6,9,56,57,58,59,60]. All manipulations were carried out at room temperature (~21 °C) unless otherwise stated. After washing in distilled water, the cells were postfixed in 1% OsO_4_ that contained 15 mg/mL K_4_[Fe(CN)_6_] and in 0.1 M sodium cacodylate buffer at room temperature for 1 h, then washed with distilled water, dehydrated through a graded series of ethanol, infiltrated with Epon 812 resin for 1 h, and polymerized at 60 °C for 24 h. Ultrathin sections were collected on carbon-coated grids with formvar support film (EMS, G2010-Cu, see Table 7), post stained with uranyl acetate and lead citrate, and viewed under an electron microscope (Tecnai 20; FEI, Eindhoven, The Netherlands).

### 4.7. Preembedding Immuno-Electron Microscopy (Pre-IEM)

Pre-IEM was performed according to He et al. [55,61], with small modifications. Briefly, after the fixation of cells with glutaraldehyde (see above), samples were washed with the blocking buffered solution (four rinses over 30 min), incubated with primary antibody dissolved in blocking solution for 4 h at room temperature, rinsed with blocking buffer (four times over 30 min), and incubated with the species-specific Fab fragments of secondary antibody labeled with 1.4 nm nanogold in blocking solution overnight at room temperature. Then cells were additionally fixed with 1.6% glutaraldehyde in 0.1M sodium cocadylate buffer (pH 7.4) for 15 min, rinsed with HEPES buffer (50 mM HEPES with 200 mM sucrose, pH 5.8, four times over 30 min), washed 3 × 5 min with PBS including glycine (20 mM sodium phosphate, pH 7.4, 150 mM NaCl, 50 mM glycine) to remove aldehydes, rinsed (3 × 5 min) with PBS–BSA–Tween (PBS containing 1% BSA and 0.05% Tween 20), and washed (3 × 5 min) with Solution E (5 mM sodium phosphate, pH 5.5, 100 mM NaCl) from the gold enhancement kit (GoldEnhance-EM 2113; Nanoprobes, Inc. see Table 7). Next, samples were placed in a mixture of manufacturer’s Solutions A and B at a 2:1 ratio (80 µL of A and 40 µL of B for 5 min and 200 µL of Solution E with 20% gum Arabic [Sigma–Aldrich. see Table 7]), and then 80 µL of Solution C was added in order to develop gold for 7–15 min. The enhancement was conducted at 4 °C. Further, samples were transferred to the neutral fixer solution composed of 250 mM sodium thiosulfate and 20 mM HEPES at pH 7.4 to stop the enhancement (three rinses over 5 min), washed with buffer E for 3-5 min, incubated in 1% OsO_4_ in 0.1 M sodium phosphate (pH 6.1) for 60 min, and rinsed with distilled H_2_O. Finally, after standard dehydrations, cells were embedded into Epon (EMS).

### 4.8. CLEM

For the identification of cells transfected with AFP, PFP, PCIII-GFP, or VFP or after cell microinjection, we used CLEM based on vertical sectioning, as described in [55,56]. For CLEM and cryo-sectioning, the cells on Petri dishes were fixed in 4% paraformaldehyde and 0.05% glutaraldehyde. The analysis of samples by two-step correlative EM tomography was performed on 200 nm-thick sections, as described previously [9,37,55]. CLEM, based on the analysis of the two-step tomographic reconstructions acquired under low (7800×) magnification with the consecutive reacquisition of the EM tomo box under high (29,000×) magnification and its re-examination, was used exactly as described [61].

### 4.9. Cryosections

Cryo-sections of cells prepared according to the Tokuyasu method as described previously [62] were immune-labeled with antibodies against albumin, GFP, PCI, VSVG, GM130, ERGIC53, GalT, and TGN38/46. For immune fluorescence, the dilution of Abs was used according to the manufacturer’s instruction. The EM dilution was 10-fold lower. Thawed cryo-sections with a thickness of 45–50 nm were labeled with an antibody (usually with the dilution 1:50) against antigens of interest and then stained with protein A conjugated with 10 nm gold particles. After contrasting with uranyl acetate, these were examined under electron microscopy (Tecnai 20; FEI/ThermoFisher Scientific, Eindhoven, The Netherland). For most of the antibodies, the dilutions used were 1:50-1:100. Block solution contained 0.1% cold-water fish-skin gelatin.

### 4.10. Electron Microscopy Tomography

Routine EM tomography and the two-step correlative EM tomography were performed as described in detail previously [2,30,62].

### 4.11. Acquisition and Statistical Analysis of the Kinetics Data

The cells were transfected, and a cell without any features of degenerative changes and with the cargo situated in the expected organelle(s) was selected for observation. The region of interest was bleached with the repetitive irradiation of the cell with a high-intensity beam or images were acquired without bleaching. When necessary, a second round of bleaching was performed. For the purpose of the economy of computer memory, images were grabbed in the format of 256 × 256 pixels. After the acquisition of the stacks of images, background and acquisition bleaching correction were performed. Briefly, using the LSM510 software, the mean fluorescence intensity was measured within the FRAP region of interest (Ifr; region of interest where FRAP occurred), the reference region of interest (Irr; within the zone of each image that was present in all images), and the base region of interest (Ibr; outside the cell of interest). In many cases, due to the shifting of the cells along the cover slip and to small changes in the shape of the GC and the cell itself during the observations, the stable and precise contouring of the Golgi complex zone was not possible, and the region of interest had to be enlarged, which made the measurements less precise.

Then the decrease in fluorescence within the cell of interest was corrected, which occured during the bleaching and before the measurement of the FRAP. The means of the corrected reference region (Icrr) were obtained by adding to each mean of the reference region (Irr) acquired after the bleaching the difference (delta) between the mean intensity just before the bleaching (Irr-pre) and that immediately after the bleaching (Irr-after): Iccr(t) = Irr(t) (1)
if variants before bleaching were used,
Iccr(t) = Irr(t) + (Irr-pre–Irr-after) (2)
if variants after bleaching were used.

Next, the actual means of the intensities within the regions of interest were estimated for each time point, with the subtraction of the background and the introduction of the correction for bleaching: Ifr-corr(t) = (Ifr(t) − Ibr(t)) * Irr-pre/Icrr(t)(3)

Two randomly selected cells from each pair (6 pairs of independent control and experiment Petri dishes; n = 6) were analyzed in each case, according to the following procedures [9]. All of the Golgi stacks were examined during the kinetics analysis. For each curve, the only variants included were those acquired every 6 s.

To convert the Y values from different datasets to a common scale, these data were normalized using the Prism software in such a way that the maximal mean of each curve was equated to 100%; namely, the mean fluorescence intensity at the time when the mean fluorescence within the region of interest was maximal (Imax) was defined as 100%. The fluorescence intensities for the other time points (In(t)) were estimated according to Equation (4): Normalized In(t) = In(t)/Imax(t) × 100(4)

To synchronize X with time, the X data were aligned versus the turning (bending) point. To determine this breakpoint, two approaches were used. According to the first approach, the point of inflection (the turning point; pivot point) was determined on the basis of the R^2^ estimation. To this end, the first two-thirds of the initial part of the curve (the plateau) was considered, and, using Excel software, the R0^2^ for the linear trend line was estimated. Then, the length of this part of the curve was increased step-by-step by consecutively adding on the next time point to the considered part of the curve, and again R1^2^, R2^2^, R3^2^, ..., Rn^2^ were estimated, and so on, for the linear trend line. Initially, this was close to 1.0. Then, when the presumed point of inflection was added, Rpt^2^ decreased. If the theoretical curve was examined after the addition of the next (i.e., after the presumed point of inflection) time point, R1^2^ then started to increase, approaching 1.0 again. If the empirical curves were assessed, the peak of this decrease was sometimes less clear. Next, the results (i.e., the consequence of Rn^2^) were plotted as a curve, and the maximal change in R^2^ was estimated for each time point. If the experimental curve did not contain a visible point of inflection, as in the case of curves that fit exponential decay, the plot of Rn^2^ did not contain any visible negative peak. The time point after which there was the maximal decrease in R^2^ (here, Rn^2^ has its maximal slope) was defined as the point of inflection (the breakpoint) and used to aligned all of the curves in such a way that this moment was simultaneous for all of them. The second approach was based on the estimation of the time point when the Golgi mass started to fragment and the spots of fluorescent begin to ‘hover’.

Thus, to align the regression lines, we calculated regression lines for the initial observation period, that is, until the end of the sampling, and then calculated regression lines for the second part of the regression curve, when a significant drop in fluorescence intensity began. The intersection of the obtained curves was considered to be the starting point for the protein exit from the Golgi zone. After compensation for bleaching and the alignment of curves using the point of bending as the common points for all regression lines, we averaged 6 curves for each series of experiments. After the normalization of the curves along the *Y*-axis and the *X*-axis, statistical analysis was performed. Using Excel software, the mean curve for each synchronization protocol was estimated and plotted with its associated standard deviation.

The datasets were also checked in terms of whether the variability of the values around the curve followed a Gaussian distribution and whether the variations in all of the curves were approximately Gaussian. The data obtained for each time point were checked using the (alpha)^2^ statistic for the distribution of six variants, because six movies for each time point were examined. Checks were also made in terms of whether any imprecision in measuring X was very small compared to the variation in Y and whether the errors were independent. The deviation of each value from the curve was seen to be random and not correlated with the deviation of the previous or next point. Paired and unpaired Student’s *t*-tests were performed to determine the significance of the differences in the comparisons of the experimental groups. All of the data are expressed as means ± SD, and the differences are considered to be significant when *p* < 0.05. For the sake of simplicity, only the biologically important meaning of a *p* < 0.05 was used for the evaluation of the statistical significance.

The kinetics curves were examined statistically for goodness of fit to exponential decay (Table 3 and Table 4). The data were checked whether these fit to a one-phase exponential curve, such that: F (t) = Fm (1 − e^−kit^), and t = ln · 2/k (5)
where k is the rate constant. However, rough examination of the normalized curves revealed that they might represent piecewise nonlinear regression when the first segment is linear (i.e., a plateau) and the second segment is an exponential decay. Several approaches were used to fit this type of model with a nonlinear regression statistical algorithm.

To align the regression lines, we calculated regression lines for the initial observation period, that is, until the end of the sampling, and then calculated regression lines for the second part of the regression curve, when a significant drop in fluorescence intensity began. The intersection of the obtained curves was considered the starting point for the protein exit from the Golgi zone. After compensation for bleaching and the alignment of curves using the point of bending as the common points for all regression lines, we averaged 6 curves for each series of experiments.

For the evaluation of the shapes of the regression curves, the following were selected: (1) exponential decay and (2) plateau followed by exponential decay, with the consecutive analysis of variance conducted for the significance of regression. To determine whether the kinetics of cargo exit from the Golgi zone decayed exponentially, several approaches were used. Initially, it was assumed that the regression curve could be viewed as distributions, and initially two empirical distributions were compared with each other, as those obtained after the empty (E) and full (F) protocols of Table 1, using Chi-squared (Chi^2^) tests. Then, the theoretical exponential distributions were compared with the empirical ones, using Chi-squared tests. Additionally, to evaluate the goodness-of-fit of the experimental curves to the theoretical exponential curves, Chi-squared goodness-of-fit was used, based on the Pearson’s chi-squared statistic and the Smirnov–Kolmogorov test, as available within the SYSTAT12 software (Version 12). Finally, using Prism software, whether the empirical regression curves were significantly different from the following regression curves was evaluated on the basis of R^2^ and Chi^2^: linear regression, exponential decay, and plateau followed by exponential decay. The *p*-values associated with the Smirnov–Kolmogorov, Fisher, and Chi-squared statistics were estimated. Furthermore, the parallelism of two curves of empirical regression was compared; namely, the curve obtained after the examination of the full Golgi complex and the curve obtained after the examination of the empty Golgi complex using Fisher statistics. The method described by Iznaga et al. [63] for linear regression was used: the regression was calculated for each curve, and two curves determined by linear regression were examined on the basis of their slopes. To do this, Y was plotted as log-transformed data, so that linear regression was suitable. The null hypothesis, H0, stated that the curves were parallel, and the alternative hypothesis, H1, that the curves were not parallel. In all cases, H0 was rejected when the difference between the normal distribution and the observed distribution was insignificant (Fd < Fst; for *p* < 005). The regression curve was considered the empirical distribution, and then using SYSTAT12 software (Version 12), the differences between the empirical and theoretical distributions were evaluated according to the Chi-squared statistic. A difference was considered significance when *p* < 0.05. Typically, eight randomly selected cells from independent experiments were assessed, with >300 movies examined. Regression curves are shown, together with the 95% confidential intervals (the green zones near the regression curves in the relevant Figures).

In each set of experiments, we used 6 such pairs of MatTek dishes (control [i.e., microinjection of an irrelevant protein or antibody or transfection with an irrelevant oligos] and experimental). Two randomly selected cells within the very center of the Petri dishes were examined. In all experiments after seeding the cells, we used 6 pairs of samples composed of one randomly selected control and one randomly selected experimental Petri dish (or the very low edges of the rat). Every pair was further processed as a single sample under completely identical conditions. After embedding into Epon, we made vertical serial 1 µm sections through the cells of interest, and then, on these vertical sections, we examined the cross diameter of the nuclei. The section where the diameter was maximal was re-embedded, and serial ultrathin vertical sections were obtained. The section where the width of our nucleus was maximal, we considered the central section, and the vertical axis was generated through the nuclear center. Ultra-sectioning was organized in such a way when control and experimental samples were embedded in one section block [64]. The control and experimental Petri dish were glued and cut as one sample. In each dish, two cells: one in the very center of the dish and another in the peripheral part of the dish, were randomly selected. Random representative images of these cells were examined. Two measurements (slices) were made in each cell. For quantification, we selected vertical sections of cells where 4 consecutive serial sections of a centriole were visible. Next, the normalized (or absolute) value for the experimental cup was calculated as a percentage of its control. These percentages were used as units (N = 6; biologically independent samples) for calculations of averages and standard deviations.

We judged the connections after the analysis of serial sections or EM tomo. Identification of procollagen I containing distensions was performed on the basis of their size on serial sections and taking into consideration the slight striation of their content. Labeling of VSVG was performed using E-nano-gold, Tokuyasu cryosections, and their gold labeling and IEM based on HRP-DAB/H_2_O_2_ reaction. Identification of the trans-side of the Golgi was performed on the basis of the presence at this side of clathrin-coated structures [38].

Endosomes were labeled with WGA conjugated with HRP. When it was necessary to label the PM, cells were incubated on ice for 20 min, and then WGA-HRP was visible on the PM. When it was necessary to labeled most endosome, TMC cells after 20 min incubation with WAG on ice were placed at 37 °C for 15 min [65]. When it was necessary to label all endosomes including TMC, the incubation time (at 37 °C) was increased to 30 min.

The discretized rotator was used for the estimation of the absolute volume [64,66]. Furthermore, we estimated the percentage of Golgi-PM carriers wherein at least two gold particles were observed as a specific labeling of GalT or Ykt6. Percentage, profile, and point counting was directly performed in the electron microscope Tecnai-20 (FEI [now ThermoFisher Scientific]). AnalySis software (ThermoFisher Scientific) was used for our measurements.

### 4.12. Statistical Analysis

Statistics were performed using GraphPad Prism 9.4.0., as described in [2]. Briefly: The number of Petri dishes used for each analysis is indicated in the figure legends, selected by a power analysis to detect a 30% change with 10% error and 95% confidence. For the sake of simplicity, Student’s *t*-tests, paired t-tests, and non-parametric Mann–Whitney U tests were used. In the majority of cases, we used a nonparametric Mann–Whitney U test. A difference was considered significant when *p* < 0.05. For a comparison of 2 groups of continuous variables with normal distribution and equal variances, 2-tailed unpaired Student’s t tests (with additional Welch correction for unequal variances) were performed with a significance threshold of *p* ≤ 0.05. In most cases, data are given as the mean ± standard deviations (SD). Values are mean ± SD of 6 variants (n = 6). In the text, the words “differ”, “smaller”, or “higher” indicate that two values are significantly (*p* < 0.05) different [67].

## Figures and Tables

**Figure 1 ijms-24-01375-f001:**
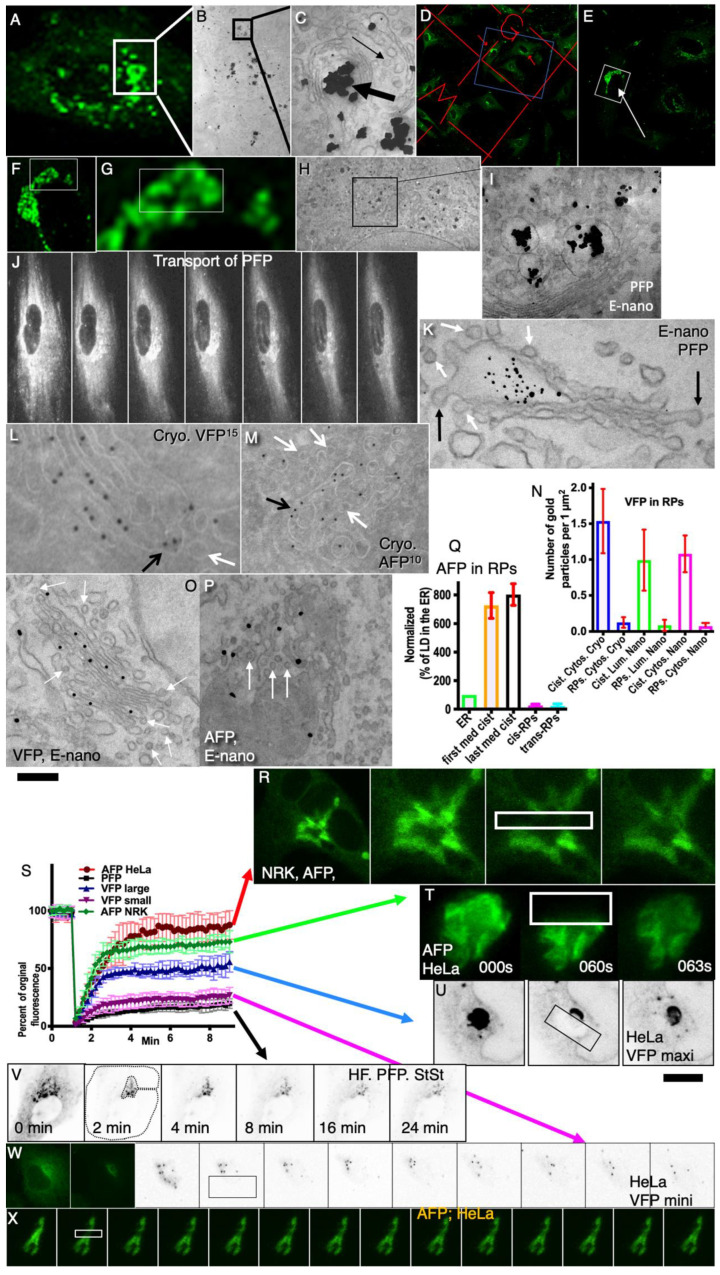
Characterization of the cargoes selected for the present study. Human fibroblasts (**A**–**K**) and HeLa cells (**L**,**M**,**O**,**P**) were transfected with PFP (**A**–**K**), VFP (**L**,**O**), or AFP (**M**,**P**), incubated at 32 °C for 16 h, and examined under immune fluorescence (**A**,**D**–**G**,**J**,**R**,**T**–**X**) or labeled with the antibody against GFP for electron microscopy (EM) (**B**,**C**,**H**,**I**,**K**–**M**,**O**,**P**). (**A**–**I**) Correlative light–electron microscopy (CLEM) of human fibroblasts transfected with PFP. At the immunofluorescence level, GFP formed dots within the Golgi zone that represented PFP-containing distensions. Each consecutive image demonstrates the elements inside the box of the previous image. (**J**) Consecutive frames of immunofluorescence show the exit of PFP from the Golgi zone (GZ). (**K**) Enhanced nanogold (E-nano). PFP is depleted in round profiles (RPs, presumably COPI-dependent vesicles; white arrows) and in COPI-coated buds (black arrow). (**L**,**M)** Cryo immunogold labeling with anti-GFP (10 nm and 15 nm gold) antibodies. VFP (**L**) and AFP (**M**) is depleted in Golgi vesicles (white arrow) but present in intercisternal connections (black arrow). (**O**,**P**) Nanogold-labeled specimens (E-nano). VFP (**O**) and AFP (**P**) are depleted in Golgi vesicles (white arrows in (**O**,**P**)). The IEM (nanogold or cryo immunolabeling) data were quantified: (**N**) concentration of VFP in Golgi RPs is lower than in Golgi cisternae independently of antibodies (Cytos and Lum). (**Q**) AFP is depleted in Golgi vesicles situated near both the first (bar: “cis-RP”) and last (bar: “trans-RP”) medial cisternae in comparison with AFP concentration in the first (bar: “firm med cist”) and the last (bar: “last med cist”) per se. (**R**–**X**) NRK cells (R) and HeLa cells (**T**,**X**) were transfected with AFP (see also Appendix A), HeLa cells with VFP (**U**,**W**), and human fibroblasts with PFP (**V**). Then, in the transfected cell, the part of the Golgi mass was bleached within the indicated boxes with the laser working at maximal intensity; images were acquired, and the ratios between the bleached areas and the areas out of the bleached boxes but within the Golgi mass were determined (see also Appendix A). (**S**) Data were plotted, and the SD was estimated. The diffusion mobility of AFP (S: green and red curves; see also Appendix A) is the highest, whereas the diffusion mobility of PFP was the lowest (S: black line). The diffusion mobility of VFP depends on the synchronization protocol used (S: blue and magenta lines; see [2]). Scale bars, 3 µm (**A**); 1.5 µm (**B**); 300 nm (**C**); (**D**) 30 µm; 12 µm (**E**,**J**); 3 µm (**F**); 1.5 µm (**G**); 750 nm (**H**) 300 nm (**I**); 155 nm (**K**,**M**); 100 nm (**L**); 280 nm (**O**,**P**); 2 µm (**R**,**T**); 10 µm (**U**–**X**).

**Figure 2 ijms-24-01375-f002:**
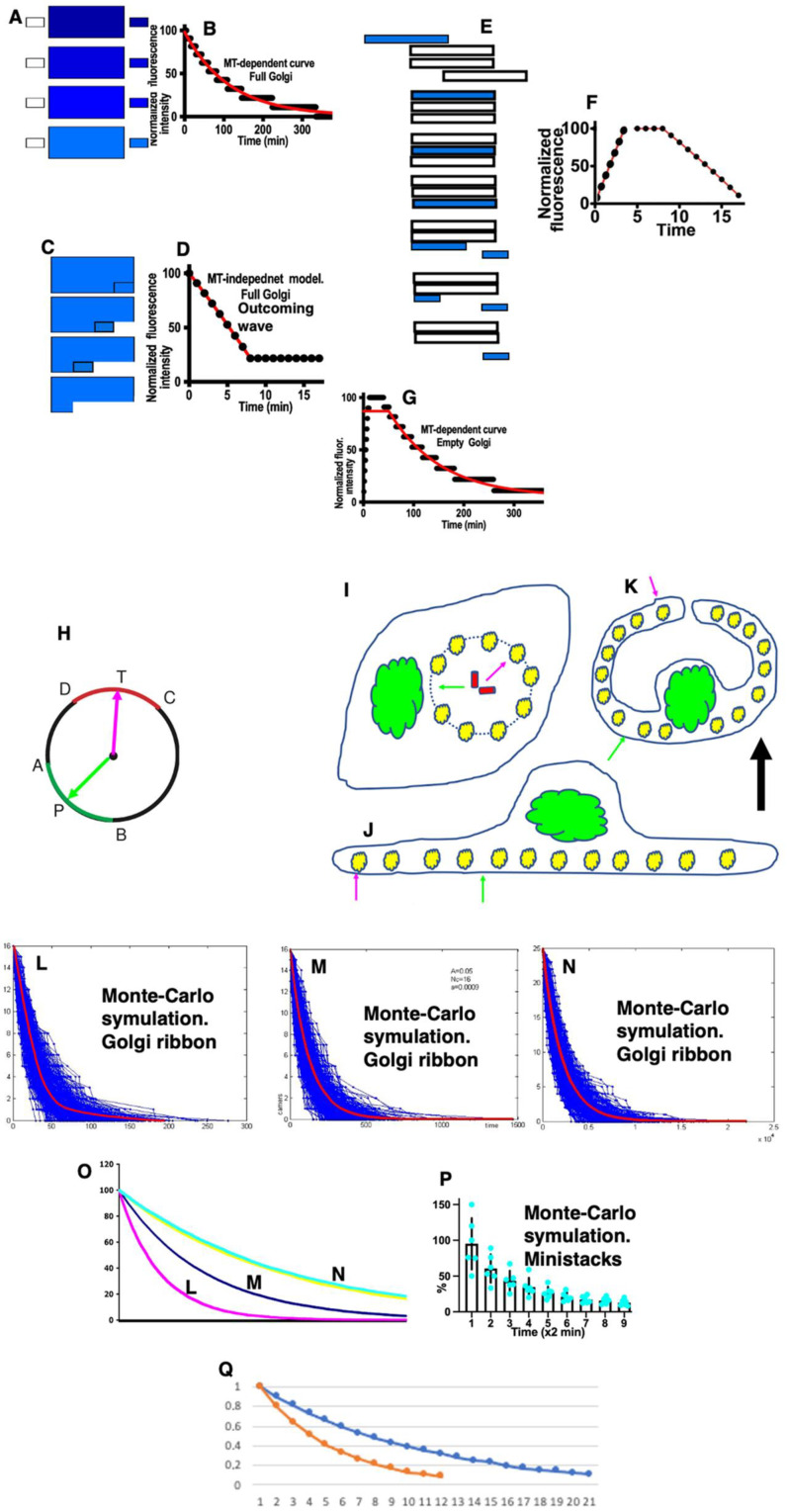
Schemes illustrating kinetics of IGT. The planels in (**A**–**G**) are explained in the text in detail. (**H**) Scheme of the dependency of the GPC departure on the microtubule growth (see Appendix A). The microtubules grow from centriole one by one. The magenta arrow illustrates the successful microtubule growth. This microtubule could hit the DTC arc in (**H**), where GPC is present. (**I**) The microtubules growing from centrioles (two red cylinders) could hit GPC (yellow structures; red arrow), inducing its departure or passing away (green arrow). For the sake of simplicity, the irregularly shaped TGN was replaced by a sphere (yellow dots). This induces the departure of the GPC present within this arc. After this, the number of GPCs and the integrated fluorescence of the remaining Golgi fragments within the GZ decreased. The regression line obtained under such conditions would be the following: y = a(1 − r)^x^, where “a” is a coefficient depending on the size of GPCs and microtubules and the speed of the microtubule growth; “r” is a coefficient determining the slope of the curve. The magenta arrow indicates the direction of the microtubule growth resulting in the its hit to post-Golgi carrier. Green arrow shows that situation when microtubules does not hit the GPC. (**J**) In cells with the fragmented GC, the possible temperature or cytoskeleton-dependent reciprocating movements are short term, and their localization is stochastic. Sometimes these cytoplasmic movements appear in the zone where mini-stacks are absent, and sometimes these movements hit mini-stacks, leading to their movement towards the PM. This would induce a situation where in the post-Golgi compartment forms contact with the PM (**J**: red [hit] and green [aside] arrows). These contacts would force SNARE-dependent fusion between the post-Golgi and the PM and disappearance of the GPC fluorescence visible in living cells. (**K**) In order to make the situation in cells with the fragmented GC similar to that in the cells with the Golgi ribbon, a flat cell could be presented as a sphere with externally acting forces directed towards the geometrical cell center. Under these conditions, the regression line would be the following: y = a(1 − r)^x^, where “a” is a coefficient depending on the size of GPCs and frequency and the intensity of the hovering of the PM and GPCs. The magenta arrow indicates the direction of the hovering of the plasma membrane resulting in the fusion between the plasma membrane and GPC. Green arrow shows that situation when the plasmalemma fluctuation occurs in the area where GPC is not present. (**L**–**P**) The Monte-Carlo simulation under different conditions produces the regression lines. (**L**–**N**) The shape of the average curves (colored in magenta) obtained during the Monte-Carlo computer simulation are similar. (**O**) Three different negatively exponential curves after their averaging (see [**L**–**N**]) were placed in one graph. The slops of these averaged lines and their declines are different. (**P**) Different types of regression lines obtained after the partial bleaching of the fragmented GC. (**Q**) Different types of exponential decay obtained after the Monte-Carlo simulation of the situation based on the fragmented GC.

**Figure 4 ijms-24-01375-f004:**
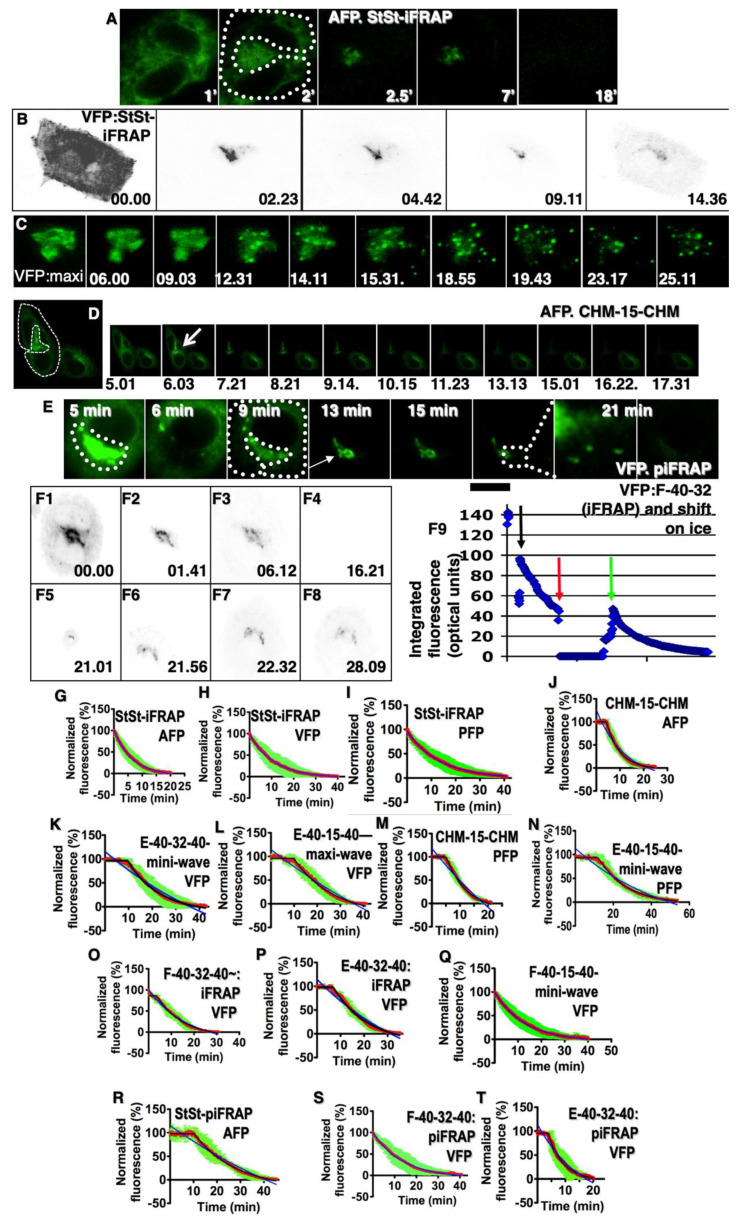
Kinetics of the cargo exit from the Golgi zone. Cells were subjected to bleaching and examined under a laser scanning confocal microscope (see Section 4). (**A**–F) Representative kymograms (the sequential regions of interest from the time-lapse recording are fused together to show dynamic changes as a two-dimensional representation) demonstrate the kinetics of a cargo exit from the GZ. (**A**) AFP. Steady state and iFRAP. (**B**) VFP. StSt-iFRAP [Table 1: 5A]. (**C**) VFP. Maxi-wave protocol (see Table 1). (**D**) AFP. CHM-15-CHM protocol (Table 1). (**E**) VFP; piFRAP protocol. (**F**) Experiment aiming to control the effect of temperature shift on the cargo kinetics. In order to test whether temperature shifts affect kinetics data, we performed control experiments. Five min after iFRAP, in the middle of the exponential curve (**F1**–**F3**), we placed the cells on ice and immediately replaced the warm medium with the cold one. The confocal microscope continued video recording (**F4**). In 10 min, we replaced the cold medium with the warm one and placed the cells back under the microscope at 32 °C. Panels (**F5**,**F6**) show the movement of the sample to the previous position. Panels (**F7**,**F8**) demonstrate that the incubation on ice stopped all transport. However, upon returning the cells to the permissive temperature under the microscope, the exponential kinetics began from the very same level within several seconds. (**F9**) The regression line describing the experiment, during which the cell after initial imaging was placed on ice (red arrow), and then the cell was returned at 32 °C, and imaging was restored. Exclusion of the time when the cells were on ice from the plot resulted in kinetics close to exponential. The kinetics of the VFP exit from the Golgi area was not significantly affected by the temperature shifts. Time points are indicated directly on the panels. (**G**–**T**) Average regression lines describing the kinetics of the cargo exit from the GZ. Protocols of cargo synchronization and the cargo used are indicated directly on graphs. Data are means ±95% confidential interval. Mini-wave protocol for AFP did not give conclusive results. (**G**–**I**) Average regression lines show that, when iFRAP protocol at steady-state was used, the average regression lines fit better to the exponential decay (see Table 3, Table 4 and Table 5). (**J**–**N**) Average regression lines demonstrate the kinetics of the cargo exit from the GZ under the conditions when the GC was empty before the re-initiation of IGT. (**J**) AFP:CHM-15-CHM. (K) VFP:mini-wave. (**L**) VFP:maxi-wave. (**M**) PFP:CHM-15-CHM. (**N**) PFP:mini-wave. (**O**–**Q**) Average regression lines demonstrate that the 40-32-40 small synchronization protocol was used the shape of the regression line depending on the time when the video recording started. When the GC was full (O:VFP), the kinetics fit to exponential decay, whereas when the GC was empty, this curve was composed of the first horizontal part, followed by the linear decay (P:VFP). Even when the mini-wave (40-15-40) protocol was used at a later stage of Golgi filling, the curve was exponential (Q:VFP). (**R**–**T**) Average regression lines describing the piFRAP protocol. (**R**) For AFP (steady state-piFRAP), the regression contains the initial horizontal part and exponential decay. (**S**,**T**) When the 40-32-40 protocols were used, the shape of the regression line also depends on the initial state (empty [**E**] of not [**F**]) of the GC. Scale bars: 5 µm (**A**,**B**,**D**–**F**); 3 µm (**C**).

**Figure 5 ijms-24-01375-f005:**
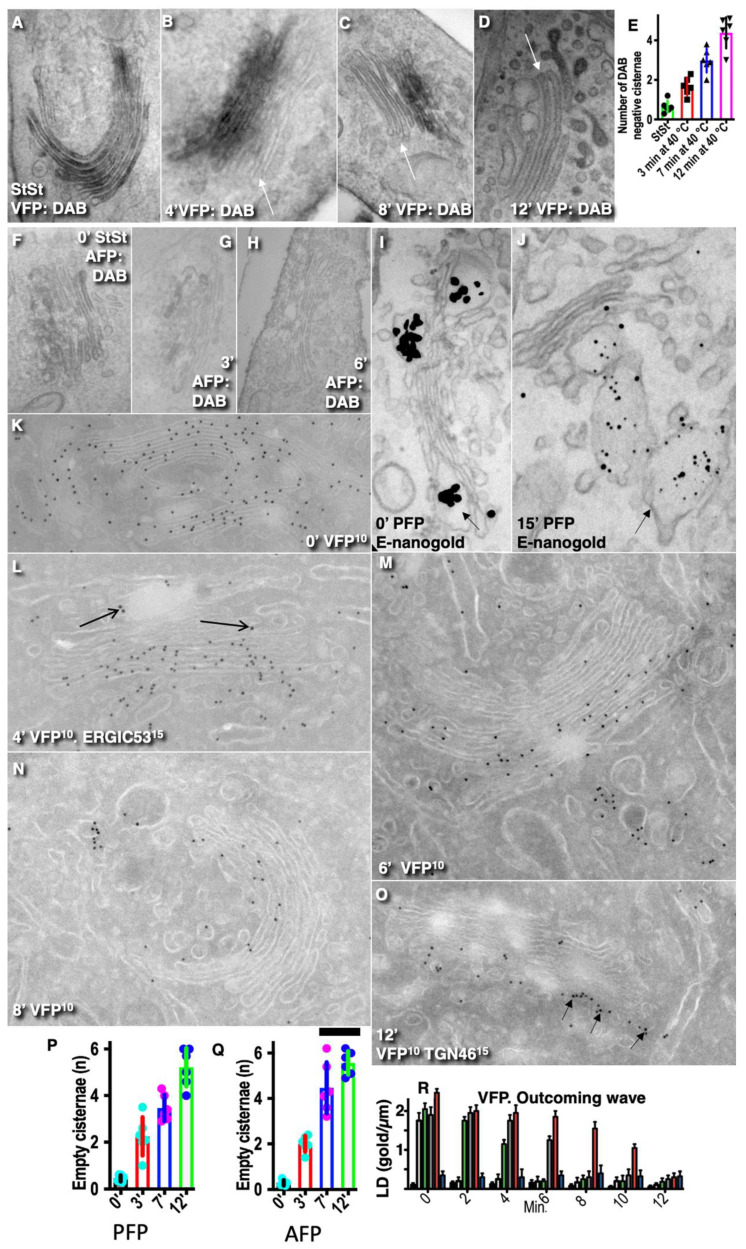
There is no equilibration of cargos across Golgi cisternae during the outgoing transport wave. (**A**–**D**; quantified in [**E**]) HeLa cells transfected with VFP were incubated at 40 °C for 3 h. Next, cells were placed at 32 ° for 30 min, returned at 40 ° (transport block), and examined after 4, 6, 8, and 12 min. VFP was detected using antibody against luminal portion of VSVG and biochemical reaction based on the application of the secondary antibody conjugated with HRP and their subsequent incubation with DAB. Time points are indicated on images. White arrows show the empty Golgi cisternae. (**E**) Dynamics of the emptying of the GC from VFP. After 7 and 12 min, the number of the DAB-negative Golgi cisternae is significantly (*p* < 0.05) higher than before the outgoing wave (StSt). (**F**–**H**) Emptying of the GC filled with AFP at steady state (quantified in **Q**). The delivery of AFP was blocked with CHM. AFP was detected using antibody against GFP and biochemical reaction based on the application of the secondary antibody conjugated with HRP and their subsequent incubation with DAB. Time points are indicated on images. (**I**,**J**) Human fibroblasts transfected with PFP were subjected to the ER accumulation–chase protocol (quantified in [**P**]). After 30 min, the GC was completely filled with PFP-positive distensions. Then the ascorbic acid was eliminated, and cells were placed at 40 °C in order to block the delivery of new portions of PFP at the Golgi complex. (**I**) Before this shift (0 min), all Golgi cisternae contained PFP distensions. (**J**) After 15 min, PFP distension was observed only within TGN, and no distensions were visible within the cisternae. Arrow indicates the post-Golgi carrier filled with PFP. (**K**–**O**) The outgoing wave for VFP examined on the basis of cryosections (quantified in [**R**]). (**K**) Starting point when VFP is present on all cisternae. (**L**) Already after 4 min, a few Golgi cisternae at the *cis*-side of the stack did not contain VFP. Arrows indicate labelling for ERGIC53. (**M**) After 6 min, a significant portion of Golgi cisternae do not contain VFP. After 8 (**N**) and 12 min (**O**), VFP was present only at the trans-side of the stack or in TMC labeled for TGN46 (arrows in [**O**]). No equilibration of VFP within different Golgi cisternae. (**P**,**Q**) Dynamics of the emptying of Golgi cisternae after synchronization of PFP (P) and AFP (**Q**) correspondingly according to the outgoing wave protocol. After 7 min and 12 min, the number of the DAB-negative (see [**F**–**H**]) Golgi cisternae is significantly (*p* < 0.05) higher than before the outgoing wave (0’). Similarly, the number of cisternae not containing PFP-positive distensions is significantly (*p* < 0.05) higher than before the outgoing wave (0’). (**R**) Initially, the GC was filled. Then the delivery of VFP was blocked, and the labeling density of VFP was measured on Golgi cisternae after different times (see graph). Initially all cisternae were filled with VFP. After 6 min, the medial cisternae at the cis-side of the stack became empty: their labeling density became significantly (*p* < 0.05) lower than at times 0’ and 2’. Scale bars: 200 nm (**A**–**D**,**F**–**H**); 420 nm (**K**); 320 nm (**L**–**O**).

**Figure 6 ijms-24-01375-f006:**
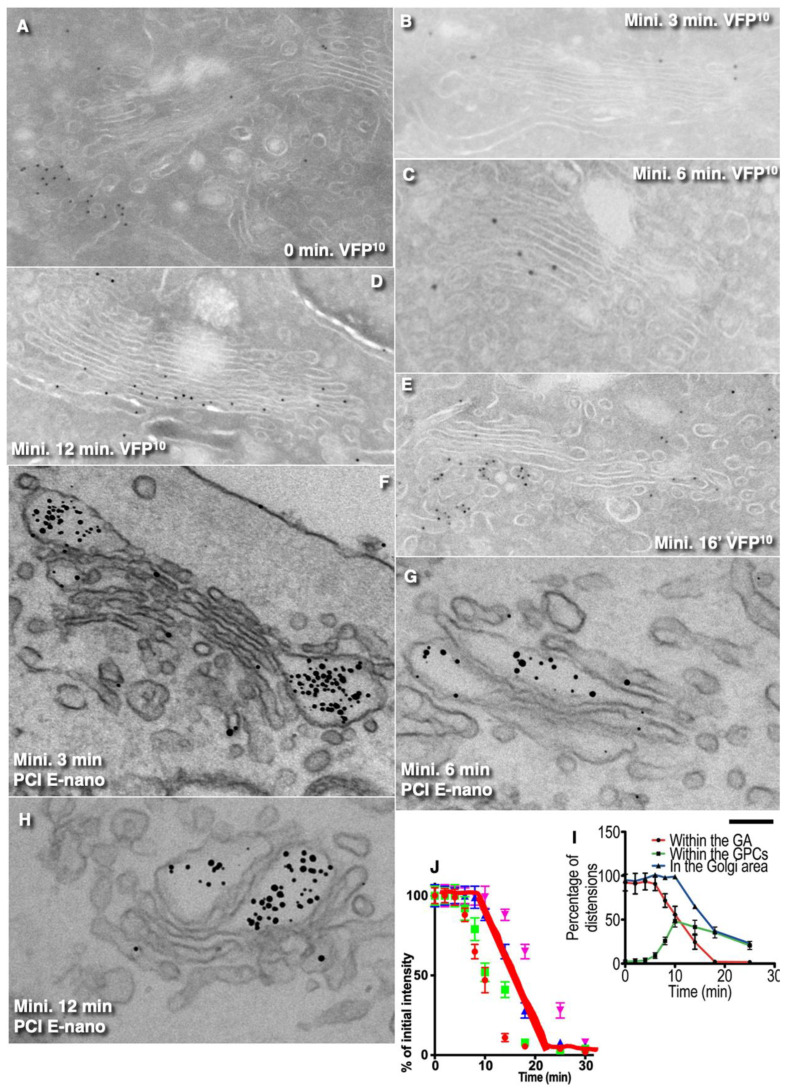
Kinetics of the cargo progression and its exit from the GC per se. HeLa cells (**A**–**E**) were transfected with VFP and subjected to the E-40-15-40-small (Table 1, 1A). After release of the 15 °C temperature block, cells were incubated at 40 °C for 3′ (**B**); 6′ (**C**); 12′ (**D**); 16 min (**E**) or were examined immediately after the first 15 °C temperature block (**A**). Cell were fixed and prepared for immuno-electron microscopy and cryo immunogold labeling with anti-GFP (10 nm gold) antibodies. Human fibroblasts (**F**–**H**) were transfected with PFP, subjected to the mini-wave (the E-40-15-40-small) synchronization protocol (Table 1, 1A), and examined using cryosection-based IEM 3 (**F**), 6 (**G**), and 12 min (**H**) after the release of the transport block. Nanogold-labeled specimens (E-nano). (**I**,**J**) Graph shows that the exit from the GC per se is linear (red line; see also Table 6). Scale bars: 250 nm (**A**–**D**,**G**,**H**); 320 nm (**E**,**F**).

**Figure 7 ijms-24-01375-f007:**
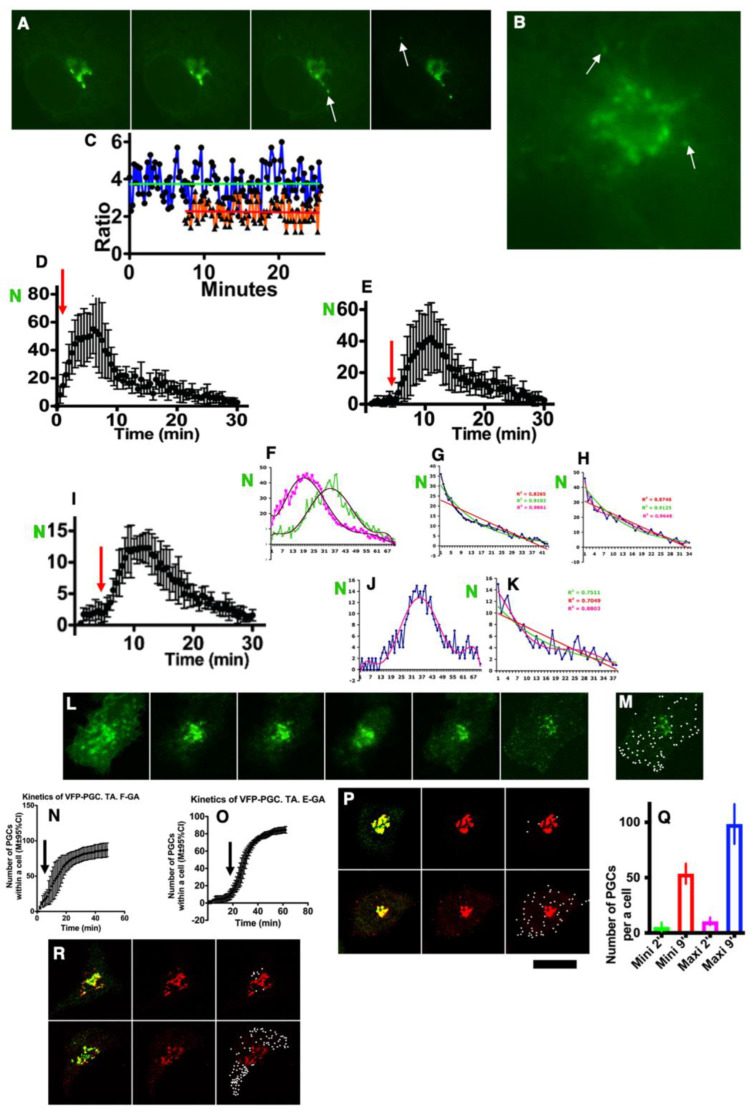
Kinetics of Golgi-to-PM carriers (see Figure 2A). (**A**) HeLa cells were transfected with VFP and then subjected to the outgoing wave protocol (steady state at 32 °C for 16 h with consecutive bleaching of the entire cell less the GC). (**B**) The GPCs (white arrows) after their departure from the post-Golgi area. (**C**) The fluorescence intensity of GPCs (ratio versus background) was measured after synchronization according to the outgoing wave (blue regression line), maxi-wave synchronization protocol (red regression line), and consecutive bleaching of the cell minus the Golgi. In both cases, the intensity of GPCs was not changed significantly (*p* < 0.05) during our observation (**B**). (**D**,**E**,**I**) The number of GPCs departing from the GC with different synchronization protocols (the outgoing wave (**D**): the Golgi was filled with VFP at pseudo-steady state (Table 2: item 3B), then the whole cell less the GC was bleached; the empty Golgi, through which the large (maxi-wave; [**D**]) or small amount of VFP was moving (mini-wave [I]). (**F**) Two regression lines are shown. Magenta is the line when the GC was filled with a cargo. The green line shows the situation when the GC was empty. (**J**) The kinetics of GPC during the maxi-wave protocol. The number of GPCs was counted every 30 sec and plotted (**F**,**J**). (**G**,**H**,**K**) The best-fit regression lines for the second parts (decays) of the kinetic curves were estimated. These parts are similar to exponential decay. Red arrows indicate the beginning of massive appearance of post-Golgi carriers. (**L**–**R**) In order to block the delivery of additional amount of cargo at the GC, we combined the piFRAP protocol with the return of cells at 40 °C. (**L**,**M**) After the iFRAP protocol and treatment of cells with tannic acid to block the GPC fusion with the PM, GPCs left the GZ but did not fuse with the PM (**M**). (**N**,**O)** The regression lines for the outgoing wave (**N**) and the maxi-wave (**O**) were estimated and plotted. Arrow indicates the beginning of the accumulation of GPCs. (**P**) When the mini-wave synchronization protocol was applied, the number of GPCs accumulated during the first 2 min was minimal (P: upper row; quantified in (**Q**): Mini 2′ [green bar]). Similarly, when the maxi-wave protocol was applied, the number of GPCs accumulated during the first 2 min was minimal (**R**: upper row; quantified in (**Q**): Maxi 2′ [magenta bar]). In contrast, the GPC number was much higher when GPCs were accumulated for 9 min (P: low row; Q, R; S: Mini 9′ [red bar] and Maxi 9′ [blue bar]). When the mini-wave protocol was applied, the number of GPCs after 9 min was significantly (*p* < 0.05) higher than after two min. Furthermore, when the maxi-wave synchronization protocol was applied, the number of PGCs after 9 min was significantly (*p* < 0.05) higher than after 2 min and also after 9 min during the mini-wave. Scale bars: 7 µm (**A**); 4 µm (**B**); 10 µm (**L**,**M**,**P**,**R**).

**Figure 8 ijms-24-01375-f008:**
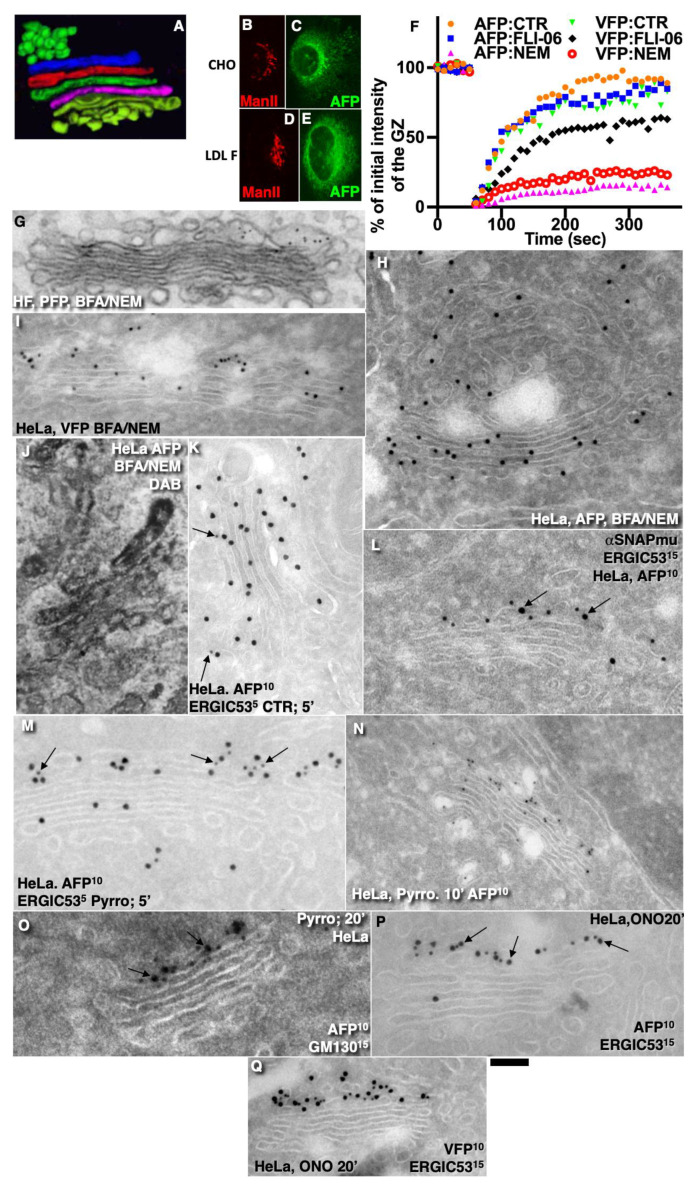
Functional coatomer I, membrane fusion, and stability of intercisternal tubules are important for IGT. (**A**) Three-dimensional model of the Golgi ministack transporting VFP. In the transporting Golgi stack, all Golgi cisternae (different colors) are connected. (**B**–**E**) Both control CHO cells (**B**,**C**) and ldl F cells (**D**,**E**) were heated at 40 °C for 2 min to block the ability of COPI in ldl F cells to form vesicles [11,22]. In the heated CHO cells, concentration of AFP within the GC increased (**C**), whereas in ldl F cells, this enrichment was lower (**E**). Green color indicates labelling for AFP. Red color indicates ManII. (**F**) Role of COPII, COPI, and SNAREs during IGT. Fluorescence intensity of the bleached Golgi area in cells transfected with AFP or VFP synchronized according to the iFRAP protocol was plotted. Orange circles: AFP:CTR (control: no treatment with inhibitors). Blue squares: cells transfected with AFP under the action of FLI-06. Magenta triangles: treatment with BFA/NEM of the AFP transfected cells. Green triangles: the VFP-transfected cells without application of the inhibitors. Black rhombs: the VFP transfected cells treated with FLI-06 (the inhibitor of COPII; see [24]). Red rings: the VFP transfected cells treated with BFA/NEM. Blockage of membrane fusion (BFA/NEM) significantly (*p* < 0.05) blocked the concentrating of AFP and VFP within the GC, whereas inhibition of the COPII function with FLI-06 did not affect significantly (*p* > 0.05) this process. (**G**) Human fibroblasts (HF) were transfected with PFP and treated with BFA/NEM-inbhibited IGT of PFP containing distensions. This treatment had lower effect on IGT in HeLA cells of AFP (**H**,**J**) and VFP (**I**). (**K**–**Q**) Impairment of the Golgi tubule stability blocked IGT of all cargoes. (**K**) Control: injection of HeLa cells transfected with AFP with an irrelevant protein. (**L**) Microinjection of the AFP transfected cells with α-SNAP mutant inhibited delivery of AFP at the trans-side of the GC after AFP synchronization according to the CHM-15-CHM protocol (compare with [**K**]). Arrows show labeling for ERGIC53 (*cis*-Golgi marker). (**M**–**O**) In HeLa cells transfected with AFP, the pyrrophenone (the inhibitor of cPLA2α) treatment inhibited the delivery of AFP at the *trans*-side of the GC. Times of treatment (5’, 10’, 20’) are shown directly on panels. Arrows show labeling for *cis*-Golgi markers: ERGIC53 (**M**) and for GM130 (**O**). (**P**,**Q**) Treatment of HeLa transfected (AFP and VFP) cells with ONO, another inhibitor of cPLA2α blocked delivery of AFP (**P**) or VFP (**Q**) at the *trans*-side of the GC in the correspondingly transfected cells. Features of experimental designs were indicated in plates. Scale bars: 180 nm (**A**); 6 µm (**C**–**E**); 350 nm (**G**–**Q**).

**Table 1 ijms-24-01375-t001:** Protocols for cargo synchronization during intra-Golgi transport.

Name of Protocol	State of the Golgi Complex (GC)	Steps of Cargo Synchronization			
		Accumulation of Cargo in ER and Emptying of the GC	Duration of Cargo Pulse	Block of Delivery of Visible Cargoes to the GC	Beginning of Video Recording
1. The small pulse (40-15-40) protocol					
A. E-40-15-40-small (mini-wave)	Empty	16 h at 40 °C	15 min at 15 °C *	Shift back to 40 °C **.	2 min after shift to 40 °C
B. F-40-15-40-small (mini-wave)	Full				10 min after shift to 40 °C
2. The large pulse (40-15-40) protocol					
A. E-40-15-40-large (maxi-wave)	Empty	16 h at 40 °C	2 h at 15 °C *	Shift back to 40 °C **.	2 min after shift to 40 °C
B. F-40-15-40-large (maxi-wave)	Full				10 min after shift to 40 °C
3. The pulse (40-32-40) protocol					
A. E-40-32-40-small	Empty	16 h at 40 °C	5 min at 32 °C	Shift back to 40 °C **.	2 min after shift to 40 °C
B. F-40-32-40-small	Full		5 min at 32 °C		10 min after shift to 40 °C
4. The emptying-pulse [32(CHM)-15-32(CHM)] protocol					
A. E-CHM-15-CHM	Empty	16 h at 40 °C, then treatment with CHM for 3 h at 32 °C	2 h at 15 °C (in absence of CHM)	Re-addition of CHM, shift back to 32 °C**.	2 min after shift to 40 °C
B. F-CHM-15-CHM	Full				10 min after shift to 40 °C
5. The iFRAP-steady-state protocol					
A. StSt-iFRAP	Full	16 h at 40 °C.	No	Bleaching of whole cell less Golgi zone	Immediately after bleaching
B. StSt-piFRAP	Full ***		3 min		
6. The pulse–outgoing wave protocol					
A. F-40-32-40 outgoing	Full	16 h at 40 °C	30 min at 32 °C	Shift to 40 °C for 30 min	By 0, 2, 4, 6, 8, 10, 12 min after shift

ER/IC, endoplasmic reticulum/intermediate compartment; iFRAP, inverse fluorescence recovery after photobleaching; piFRAP, photobleaching inverse fluorescence recovery after photobleaching. * If PFP undergoes synchronization, ascorbic acid (AA) was added to the medium to allow PFP exit from the ER, whereas, after the shift to 40 °C, AA was eliminated from the medium. ** Bleaching of whole cell minus the Golgi area was conditional. *** After bleaching of the Golgi complex, it remained filled with cargo, but it did not contain visible cargoes.

**Table 2 ijms-24-01375-t002:** Rates of cargo diffusion along the Golgi ribbon (half-time in minutes: mean ± SD).

Cargo Protein	When FRAP Examined, after Release of a Transport Block	Rates of Cargo Diffusion through Golgi (Half-Time; Min; Mean ± SD), According to Amount of Cargo Transported	
		Small	Large
VFP	Early	6.0 ± 0.9	1.4 ± 0.6
	Late	5.5 ± 1.1	4.9 ± 1.0
PFP	Early	Undetectable	Undetectable
	Late	Undetectable	Undetectable
AFP	Early	0.3 ± 0.05	ND
	Late	1.1 ± 0.07	ND

ND, not determined. Cells were processed to measure FRAP as described in Methods. Half-time was measured using graphs of FRAP.

**Table 3 ijms-24-01375-t003:** Evaluation of empirical curves for the goodness-of-fit of the theoretical curves to exponential decay (see also Appendix A).

Cargo Protein	Protocol	Significance of Statistic						
		DEL	CST	CSGF	CST (Chi^2^)	SKT	R^2^	
							ED	PED
VFP	1A. E-40-15-40-small	*	*	*	*	*	0.851	0.932
	1B. F-40-15-40-small	*	ns	ns	ns	ns	0.943	0.941
	2A. E-40-15-40-large	*	*	*	*	*	0.812	0.950
	2B. F-40-15-40-large	*	ns	ns	ns	ns	0.934	0.923
	3A. E-40-32-40-small	*	ns	ns	ns	ns	0.879	0.935
	3B. F-40-32-40-small	*	*	*	*	*	0.937	0.961
	5A. E-40-32 (iFRAP)	ND	*	*	*	*	0.836	0.949
	5B. E-40-32 (piFRAP)	ND	*	*	*	*	0.853	0.954
	5C. F-40-32 (iFRAP)	ND	ns	ns	ns	ns	0.945	0.929
	5D. F-40-32 (piFRAP)	ND	ns	ns	ns	ns	0.928	0.927
	6A. StSt-iFRAP	ND	ns	ns	*	*	0.921	0.911
	7A. F-32-40 outgoing	ND	ns	ns	ns	ns	0.931	0.933
AFP	4A. E-CHM-15-CHM	*	*	*	*	*	0.817	0.947
	4B. F-CHM-15-CHM	*	ns	ns	ns	ns	0.925	0.954
	6A. StSt-iFRAP	*	ns	ns	ns	ns	0.941	0.939
	6B. StSt-piFRAP	*	*	*	*	*	0.811	0.933
SFP	4A. E-CHM-15-CHM	*	*	*	*	*	0.818	0.951
	4B. F-CHM-15-CHM	*	ns	ns	ns	ns	0.928	0.921
PFP	1A. E-40-15-40-small	*	*	*	*	*	0.826	0.947
	1B. F-40-15-40-small	*	ns	ns	ns	ns	0.939	0.931
	2A. E-40-15-40-large	*	*	*	*	*	0.841	0.943
	2B. F-40-15-40-large	*	ns	ns	ns	ns	0.942	0.939
	6A. StSt-iFRAP	ND	ns	ns	ns	ns	0.933	0.929

The H0 hypothesis assumed that the empirical distribution belongs to the theoretical one, which was rejected when the difference between the theoretical distribution and observed distribution was insignificant (Fd < Fst for *p* < 0.05); None of the empirical curves of regression fit the linear regression; All of the empirical curves obtained under the conditions when the Golgi complex was ribbon-like followed the regression curve with a plateau followed by exponential decay.; ns, not significant; *p* > 0.05; *, *p* < 0.05; ND, not determined; DEL, difference between two empirical regression curves; CSGF, Chi-squared goodness-of-fit; CST, Chi-squared test; SKT, the Smirnov–Kolmogorov test; R^2^, Coefficient of determination or R-squared test; ED, exponential decay: comparison between the observed distribution and exponential decay; PED, plateau followed by exponential decay: comparison between the observed distribution and the plateau followed by exponential decay.

**Table 4 ijms-24-01375-t004:** Statistical analysis for goodness-of-fit of regression curves that describe the kinetics of cargo exit from the Golgi zone.

Protocol	Regression Curves		
	AFP	PFP	VFP
1A. E-40-15-40-small	-	PED	PED
1B. F-40-15-40-small	-	ED	ED
2A. E-40-15-40-large	-	PED	PED
2B. F-40-15-40-large	-	ED	ED
3A. E-40-32-40-small	-	-	PED
3B. F-40-32-40-small	-	-	ED
4A. E-CHM-15-CHM	PED	-	-
4B. F-CHM-15-CHM	ED	-	ED
5A. E-40-32 (iFRAP)	-	-	PED
5B. E-40-32 (piFRAP)	-	-	PED
5C. F-40-32 (iFRAP)	-	-	ED
5D. F-40-32 (piFRAP)	-	-	ED
6A. StSt-iFRAP	ED	ED	ED
6B. StSt-piFRAP	PED	-	-
7A. F-32-40 outgoing	-	-	ED

ED, exponential decay has highest goodness-to-fit. PED, plateau, followed by exponential decay, has highest goodness-to-fit.

**Table 5 ijms-24-01375-t005:** Statistical evaluation of the goodness-of-fit of the regression lines obtained after synchronization of VFP and PFP exit from GZ according to the piFRAP protocol.

Theoretical Curves	Cargoes			
	VFP		PFP	
	R Square (R^2^)	Absolute Sum of Squares (Chi^2^)	R Square (R^2^)	Absolute Sum of Squares (Chi^2^)
1. A linear decay.	0.851 ± 0.21	204’311 ± 35,103	0.832 ± 0.12	195’415 ± 45,216
2. An exponential decay.	0.884 ± 0.13	182’365 ± 29,010	0.864 ± 0.20	162’461 ± 38,212
3. The horisontal plateau followed by an exponential decay.	0.922 ± 0.23	124’035 ± 11,102	0.964 ± 0.27	49’561 ± 19,214
4. A linear decay followed by an exponential decay.	0.971–0.013	52’615 ± 23,114	0.910 ± 0.19	134’056 ± 12,203
5. The horisontal plateau followed by the linear decay.	0.824 ± 0.023	146’035 ± 15,502	0.824 ± 0.27	397’662 ± 20,254

**Table 6 ijms-24-01375-t006:** Statistical evaluation of the goodness-of-fit of the regression lines obtained after synchronization of VFP and PFP exit from the GC per se.

Theoretical Curves	Cargoes			
	VFP		PFP	
	R Square (R^2^)	Absolute Sum of Squares (Chi^2^)	R Square (R^2^)	Absolute Sum of Squares (Chi^2^)
1. A linear decay.	0.863 ± 0.29	231’311 ± 46,103	0.829 ± 0.32	195’415 ± 39,277
2. An exponential decay.	0.828 ± 0.18	199’305 ± 29,010	0.833 ± 0.29	158’476 ± 34,416
3. The horisontal plateau followed by an exponential decay.	0.833–0.13	336’615 ± 27,004	0.799 ± 0.26	123’705 ± 14,445
4. A linear decay followed by an exponential decay.	0.873 + 0.43	275’333 ± 30,405	0.843 ± 0.34	211’112 ± 41,005
5. The horisontal plateau followed by the linear decay.	0.974 ± 0.23	33’034 ± 11,702	0.955 ± 0.32	38’963 ± 18,018

**Table 7 ijms-24-01375-t007:** Reagents.

Reagent or Resource	Source	Identifier (Catalog no)
Antibodies		
Rabbit polyclonal αnti-GFP (EM dilution 1:100)	Sigma-Aldrich, Milan, Italy	SAB4301138
Goat polyclonal α-mouse IgG-Cy3 (WB 1:500)	Sigma-Aldrich, Milan, Italy	AP124C
Goat polyclonal α-mouse IgG-Cy2 (WB 1:1000)	Abcam, Cambridge, UK	ab97034
Goat polyclonal α-rabbit IgG-Cy2 (WB 1:1000)	Abcam, Cambridge, UK	ab6944
Goat polyclonal α-rabbit IgG-Cy3 (WB 1:1000)	Abcam, Cambridge, UK	ab6939
Antibody secondary Alexa 488, 566 and 633	Sigma-Aldrich, Milan, Italy	
Rabbit polyclonal antibodies against GFP	Abcam, Cambridge, UK	ab6556
Monoclonal antibody against the luminal domain of VSVG	Dr. J. Grunberg.	N/A
Mouse monoclonal antibody against the luminal domain of VSVG VSV-G (F-6)	Santa Cruz Biotechnology, Dallas, TX, USA	sc-365019
The I-14 monoclonal antibody against folded VSVG	D.S. Lyles (Wake Forest University School of Medicine)	N/A
Rabbit polyclonal antibody against GS15	MyBioSource, Inc.	MBS2525056
Rabbit polyclonal antibodies against the luminal domain of VSVG	Dr. K. Simons.	N/A
Invitrogen rabbit polyclonal antibody against YKT6	ThermoFisher Scientific	PA5-112789
Mouse Anti-VSV Recombinant Antibody (CBMJV-0164)	Creative Biolabs, Shirley, New York, NY, USA	CBMAB-V208-0167-WJ
Rabbit polyclonal antibody against TGN46	ThermoFisher Scientific, Milan, Italy	PA5-23068
Invitrogen rabbit polyclonal antibody against Transferrin Receptor	ThermoFisher Scientific, Milan, Italy	PA5-83022
Rabbit polyclonal antibody against VAMP3	ThermoFisher Scientific, Milan, Italy	PA5-116218
Chemicals, peptides, and recombinant proteins		
Epon812	Sigma-Aldrich, Milan, Italy	25134-21-8
Cycloheximide	Sigma-Aldrich, Milan, Italy	01810
GlutaMAX™ Supplement	Thermo Fisher Scientific, Milan, Italy	35050061
Uranyl acetate	Serva, Catoosa, OK, USA	6159-44-0
GoldEnhance EM	Nanoprobes, Yaphank, NY, USA	2114
HQ Silver	Nanoprobes, Yaphank, NY, USA	2012-45ML
Fugene 6 transfection reagent	Roche, Indianapolis, IN, USA.	N/A Purchased in 2009
MatTek chambers	Nalge Nunc International, Naperville, IL, USA	P35G-1.5-14-CGRD
CO_2_ independent medium	Invitrogen, Carlsbad, CA, USA	
DPBS with D-glucose and sodium pyruvate	Invitrogen, Waltham, MA, USA	14287-080.
Serum and amino acid-free Earle’s balanced salt solution	Gibco (Now Thermo Fisher Scientific).	24010043
Dulbecco’s modified Eagle’s medium/F12	ThermoFisher Scientific, Milan, Italy	1320033
Fetal bovine serum	Sigma-Aldrich, Milan, Italy	F7524
Pyrrophenone	Santa Cruz Biotechnology Dallas, TX, USA	341973-06-6
Penicillin	Sigma-Aldrich, Milan, Italy	1504489
Streptomycin	Sigma-Aldrich, Milan, Italy	3810-74-0
PEG with the molecular mass of 1500-2000 MW	Sigma-Aldrich, Milan, Italy	25322-68-3
WGA-HRP	Vector Laboratories Newark, CA, USA	PL-1026-2

## Data Availability

Not applicable.

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
