# Peer review of "The Diffusion Model of Intra-Golgi Transport Has Limited Power"

_ijms, 2023, doi:10.3390/ijms24021375_

Round 1
Reviewer 1 Report
In this manuscript, using immunofluorescence and electron microscopy, the authors assessed the kinetics of a variety of inducible soluble and transmembrane cargoes GFP-tagged albumin, collagen I and VSVG) trafficked through the Golgi. They confirm the absence of cargo in vesicle-like profiles and are consistent with their previous publications, showing the presence of cargo molecules only within the lumen of the Golgi. This manuscript appears to be a follow-up of the authors’ previous publication, where they make an argument in support of the (kiss-and-run) KARM model. Therefore, it seems as if they are trying to further support the KARM model by reporting limitations of the diffusion model (DM). This study does not present much new data but makes a lot of reinterpretations. This paper is not very different from Beznoussenko et al (PMID: 24867214) work which found that the transport kinetics of the same soluble cargo favored the diffusion model.
The major finding of this paper is the transport of soluble cargo upon chemically abolishing intracisternal connections using NEM+BFA treatment. While NEM does block NSF and thereby prevent membrane fusion, NEM is highly reactive and targets a wide range of biochemical processes. It would be more appropriate for the authors to use the Tet-inducible dominant NSF E329Q mutant.
Importantly, this paper is poorly presented. Apart from the numerous typos and punctuation errors, the authors need to significantly improve the figure legends (maintain and alphabetical order), list of abbreviations etc.
Major issues:
1. The texts are disorganized to understand the results and the rationale in support of applying the experiments. As an example, the text section of Figure 1 has been described in figure 6. Even for a specialist, it was very difficult to keep track while reading the paper.
2. Table 2 (and Figure 3) are very difficult to understand. The rationale for different incubations and methods should be explained upfront with sufficient details.
3. It would be essential to use a Golgi marker (for instance, GM130) to measure the kinetics of cargo exit from the Golgi zones when different synchronization protocols were used.
4. The limitation of this study is not using alternative “cargo-wave” approaches, like the RUSH approach, that allows more “physiological” cargo, specifically for glycosylated and transmembrane proteins. VSVG is an overexpressed viral protein that can induce membrane fusion and albumin is an unusual soluble cargo lacking glycosylation.
5. Figure legends are not written in an organized manner; thus, it becomes challenging to understand what can be predicted from each figure.
6. We could not find any data in the paper supporting the statement that PFP (procollagen I N-terminally tagged with GFP, which should be abbreviated GFP-PCI) is functional, as the authors claimed.
7. It should be mentioned that VSVG-GFP is actually a temperature-sensitive VSVG mutant.
8. Scientific American (ref 1) is not a peer-review publication.
9. Overall, there are a lot of missing parts in the texts, such as in figure 7, which is not mentioned anywhere (both text and figure legends) that electron microscopy has been performed.
Author Response
We corrected the manuscript, moved Figure 6, graphs and the model schemes backward to the beginning of Results, inserted videos 5 and 6. The sixth video demonstrates a tomo of five combined serial tomo-boxes of the transporting mini stack. We corrected tables withR2 and Chi2.
Reviewer 1
- Moderate English changes required.
Reply: We corrected our English.
- In this manuscript, using immunofluorescence and electron microscopy, the authors assessed the kinetics of a variety of inducible soluble and transmembrane cargoes GFP-tagged albumin, collagen I and VSVG) trafficked through the Golgi. They confirm the absence of cargo in vesicle-like profiles and are consistent with their previous publications, showing the presence of cargo molecules only within the lumen of the Golgi. This manuscript appears to be a follow-up of the authors’ previous publication, where they make an argument in support of the (kiss-and-run) KARM model. Therefore, it seems as if they are trying to further support the KARM model by reporting limitations of the diffusion model (DM).
Reply: Indeed, we tried to prove that existing models of intracellular transport namely, the vesicular model, the diffusional model and the cisternae maturation-progression models are less powerful than the KAR model. The application of the KARM for intra-Golgi transport was proposed by us in 2008 (The Golgi Apparatus. State of the art 110 years after Camillo Golgi’s discovery (Eds: Mironov A.A. and Pavelka M.). Chapter 3.2. Wien. Springer-Verlag, pp. 342-357. In 2012 (doi:10.3390/ijms130x000x), we published the paper showing that this model is compatible with mathematical modelling of this process. In 2013 (doi: 10.1007/s00418-013-1141-6), we demonstrated that the KARM is compatible with many experimental data. In 2014 (doi: 10.7554/eLife.02009), we demonstrated that the intra-Golgi transport of albumin cannot be explained by the vesicular model and by the cisterna maturation-progression model (CMPM). In 2016 (doi: 10.1007/s00418-016-1483-y), we proved that COPII vesicles do not exist in yeast. In 2019 (doi: 10.3389/fcell.2019.00146), we demonstrated that all even very new data favoring the cisternae maturation-progression model. In 2022 (doi: 10.3390/ijms23042152), we directly compared the KARM and the CMPM and proved that the CMPM is less powerful. In the current paper, we compared the KARM and the diffusion model. The famous philosopher of science I. Lakatos I. (Falsification and the Methodology of Scientific Research Programmes. // Criticism and the growth of Knowledge. Ed. by I. Lakatos and A. Musgrave. Cambr. University Press, 1970. P. 91–195) demonstrated that the discovery of new theories is always accompanied by re-interpretation of already published works. Therefore, our re-interpretations are obligatory events during our fighting in favor of the correct model of intracellular transport.
- This study does not present much new data but makes a lot of reinterpretations.
Reply: Indeed, we proposed many re-interpretations. However, these new hypotheses are based on the completely new data. For instance, Patterson et al., (2008) demonstrated that when the Golgi complex is filled with a cargo, the exit from it occurs according to the negatively exponential regression line. Our data demonstrated that these results are ad hook because Patterson et al. (2008) did not check the exit from the Golgi complex under these conditions when initially, the Golgi was the empty Golgi complex and then it is filled with a small amount of cargo. This induced a huge misinterpretation very harmful for the scientific society. Also Patterson et al. (2008) did not check the dynamics of the cargo movement through the Golgi complex per se. Finally, we indicated in the text our completely new observations.
For the first time, the kinetics of the exit of various commonly used standard cargoes from the Golgi zone was studied under conditions when the synchronization of the cargo wave begins under conditions when the GC does not contain cargo. It is proved that the kinetics of the passage of various cargoes through the GC corresponds to straight regression line.
The observations that there is no balancing of cargo concentration between different cisternae are completely original. It is proved for the first time that transport through different CG compartments requires the SNARE function, it is established that the connections between different GC cisternae are not constant, which rejects the diffusion pure model and calls into question the universality of using a negative exponential curve for cargo exiting the GC zone.
- This paper is not very different from Beznoussenko et al (PMID: 24867214) work which found that the transport kinetics of the same soluble cargo favored the diffusion model.
Reply: Indeed, our previous work demonstrated that connections between Golgi cisternae could be used for the diffusion of GFP–albumin. However, in that paper we did not disprove the data by Patterson et al (2008) who demonstrated that exit of visible cargoes from the Golgi zone occurs according to the exponential decay, which seems to support our data (Trucco et al., 2004) where we described formation of intercisternal connections when cargo synchronously moved through the Golgi complex. However, we think that the main discovery of the current paper is the explanation of the contradictions induced into the field by the paper published by Patterson et al (2008) and rejection of the diffusion model which is based mostly on our data (Trucco et al., 2004). The paper on RUSH is prepared for publication and ready for submission (see https://disk.yandex.ru/i/tMe27ccvbLiMRQ
https://disk.yandex.ru/i/Zc6QdQy37m6xtw) on the exit of different cargo proved that our previous explanation of the results describing transport of albumin was incorrect. We indicated in the text observation which were made for the first time.
- The major finding of this paper is the transport of soluble cargo upon chemically abolishing intracisternal connections using NEM+BFA treatment. While NEM does block NSF and thereby prevent membrane fusion, NEM is highly reactive and targets a wide range of biochemical processes. It would be more appropriate for the authors to use the Tet-inducible dominant NSF E329Q mutant.
Reply: We included new data based on the application of the microinjection of the His6-α-SNAP (L294A) mutant protein. The inducible systems are too slow for our experiments.
- Importantly, this paper is poorly presented. Apart from the numerous typos and punctuation errors, the authors need to significantly improve the figure legends (maintain and alphabetical order), list of abbreviations etc.
Reply: We corrected all these mistakes, re-wrote and reorganized Figures and eliminated most of abbreviations.
Major issues:
1a. The texts are disorganized to understand the results and the rationale in support of applying the experiments. As an example, the text section of Figure 1 has been described in Figure 6. Even for a specialist, it was very difficult to keep track while reading the paper.
Reply: We re-wrote the text trying to make it more understandable and added explanations in the description of these protocols.
2a. Table 2 (and Figure 3) are very difficult to understand. The rationale for different incubations and methods should be explained upfront with sufficient details.
Reply: We explained better Table 2 and Figure 3 and added some new data there.
3a. It would be essential to use a Golgi marker (for instance, GM130) to measure the kinetics of cargo exit from the Golgi zones when different synchronization protocols were used.
Reply: GM130 is not a specific Golgi marker. It is a marker of the cis-most cisternae and partially ERGIC whereas TGN38/46 is the marker of the trans-most cisterna. We used ManII as much more established marker of the medial Golgi cisternae.
4a. The limitation of this study is not using alternative “cargo-wave” approaches, like the RUSH approach, that allows more “physiological” cargo, specifically for glycosylated and transmembrane proteins. VSVG is an overexpressed viral protein that can induce membrane fusion and albumin is an unusual soluble cargo lacking glycosylation.
Reply: In our discussion we explained why we did not use in the current paper the RUSH system. Briefly, RUSH is not reversible and thus, the delivery of a cargo at the Golgi complex cannot be stopped. Importantly, the RUSH method is similar to the synchronization protocol when after incubation at 40ËšC cells were shifted to 32ËšC. RUSH does not allow to used very small and simultaneously similar small amount of cargo moving through the Golgi complex. However, our paper based on RUSH is ready for publication (see https://disk.yandex.ru/i/tMe27ccvbLiMRQ
https://disk.yandex.ru/i/Zc6QdQy37m6xtw). The regularities there are the same. On the other hand, there are more than 300 enzymes involved in glycosylation and located in the Golgi complex. Their functions overlap and duplicate each other in many ways (Varki, 1998; doi: 10.1016/s0962-8924(97)01198-7). Polysaccharide chains could be attached to asparagine (N-glycosylation) and serine/threonine (O-glycosylation) located in the polypeptide chain. However, the addition of these chains is also possible to lysine, tryptophan and tyrosine. Moreover, there are several similar sites in the RUSH polypeptide. These sites could be glycosylated (Boncompain. et al., 2012; doi: 10.1038/nmeth.1928). Moreover, when the glycosylation of proteins is disrupted due to their mutations, their transport to the plasma membrane is practically not inhibited and embryos are formed (Dennis et al., 1999; PMID: 10376012; Stanley et al., 1990, doi: 10.1007/BF01233357; Finnie and O'Shea, 1989, doi: 10.3109/00313028909061058; Ohtsubo and Marth, 2006; doi: 10.1016/j.cell.2006.08.019.). That is why glycosylation inhibitors are toxic, non-specific (Finnie and O'Shea, 1989). Moreover, RUSH is glycosylated almost like VSVG (Boncompain et al., 2012). If the inhibitors were specific, or if the removal of sites to which the polysaccharide chain is attached were specific, they would be used for the synchronization of the cargo intra-Golgi transport long time ago. But they are not used for this. When we were preparing articles by Bonfanti et al. (1998; doi: 10.1016/s0092-8674(00)81723-7) and Mironov et al. (2001; doi: 10.1083/jcb.200108073), we tried different such inhibitors for synchronization, but without success and we had to use 2,2'-dipyridyl and temperature shifts.
5a. Figure legends are not written in an organized manner; thus, it becomes challenging to understand what can be predicted from each figure.
Reply: We re-wrote all Figure legends.
6a. We could not find any data in the paper supporting the statement that PFP (procollagen I N-terminally tagged with GFP, which should be abbreviated GFP-PCI) is functional, as the authors claimed.
Reply: It is presented in Figure 1A-I based on CLEM data. Moreover we corrected the description and indicated that this chimera is functional in the transport sense. Moreover, this cargo was characterized by Patterson et al. (2008) from functional point of view.
7a. It should be mentioned that VSVG-GFP is actually a temperature-sensitive VSVG mutant.
Reply: We mentioned this (red text)
8a. Scientific American (ref 1) is not a peer-review publication.
Reply: We replaced this reference (ref. 1)
9a. Overall, there are a lot of missing parts in the texts, such as in Figure 7, which is not mentioned anywhere (both text and figure legends) that electron microscopy has been performed.
Reply: We corrected this problem in the text (yellow text).
Reviewer 2 Report
This study explores whether the diffusion model, suggested in earlier studies to be a possible trafficking model and that can co-exist with the maturation model, would actually be feasible. Authors use a variety of basic cellular biophysics and microscopy techniques to evaluate the spatio-temporal localization and behavior of the studied molecules coupled with the kinetics of entry, diffusion and exit of cargo from the Golgi apparatus.
Comments:
It would improve the manuscript, if the authors could perform a functional assay such as glycosylation of VFP and the PFP under different conditions to validate the observed trafficking behavior.
In an ealier paper (Beznoussenko et al. 2014), authors suggest that low pH in TGN could be the reason behind cargo retention and that concanamycin treatment abolished such concentration, which supports the diffusion model. How do authors explain this in the context of this manuscript?
General: It would be better for the flow of the manuscript if the English language would be checked by a native English speaker. In many instances it was difficult to read and follow.
Author Response
We corrected the manuscript, moved Figure 6, graphs and the model schemes backward to the beginning of Results, inserted videos 5 and 6. The sixth video demonstrates a tomo of five combined serial tomo-boxes of the transporting mini stack. We corrected tables withR2 and Chi2.
Reviewer 2
This study explores whether the diffusion model, suggested in earlier studies to be a possible trafficking model and that can co-exist with the maturation model, would actually be feasible. Authors use a variety of basic cellular biophysics and microscopy techniques to evaluate the spatio-temporal localization and behavior of the studied molecules coupled with the kinetics of entry, diffusion and exit of cargo from the Golgi apparatus.
Comments:
It would improve the manuscript, if the authors could perform a functional assay such as glycosylation of VFP and the PFP under different conditions to validate the observed trafficking behavior.
Reply: We performed experiments with inhibitor of glycosylation and VSVG-GFP. The kinetics was the same.
In an ealier paper (Beznoussenko et al. 2014), authors suggest that low pH in TGN could be the reason behind cargo retention and that concanamycin treatment abolished such concentration, which supports the diffusion model. How do authors explain this in the context of this manuscript?
Reply: We prepared separated paper on the exit of cargo from the Golgi and could send it to the reviewer. In any case, our previous suggestion is based on the hypothesis that there could be cisternae maturation together with the diffusion model. However our recently published paper (Beznoussenko et al., 2022) and the current paper suggest that the cisterna-maturation progression model has no significant power in explanation of experimental data.
General: It would be better for the flow of the manuscript if the English language would be checked by a native English speaker. In many instances it was difficult to read and follow.
Reply: The text was checked by the Englishman.
Round 2
Reviewer 1 Report
The authors significantly modified the manuscript, but despite their efforts, the updated text is mostly unreadable to this reviewer. It will require a complete rewrite to be accessible to the general public. The text of the article gives an impression of the second draft version, which is halfway to the high-quality work. This version's most annoying feature is multiple figure and figure legend errors.
Below are some examples:
Fig1. L,M,O,P and are misplaced or mislabeled or both.
In text:
N is for L-M
Q is for O,P. meanwhile, it is written that Fig 1M,P are quantified in 1Q
1M is discussed twice: as VFP and as AFP
L-M are VFP
O-P are AFP
R-X portion of Fig1 should be described in a logical way: NRK, HeLa, HF- the way it is presented as images.
1G, H are about PFP, not AFP.
The 1S symbol legends should be in logical order from highest to lowest
L-Q are all messed up in results, figure legend text and in images names:
In Fig1 legend:
VFP (L,M)
AFP (O,P)
But the figure is different
In Fig1 legend:
(L,O) cryo
(O,P) nano
But the figure is different
Fig2. legend represents complete duplication of the text presented in the Results section. In fact, the is some repetition of words can be found due to the careless copy/paste for Fig2I
Fig3. Not clear which cells and what label was used.
Figure legend has the same text repetition issue for A-K graphs.
L-M graphs are not discussed in the Results section at all.
Fig3U is not explained at all.
What is (C) about?
Scale bar does not exist, even outside of the images as on Fig1. Such random placement is hard to understand.
2.2 Results
Second part of Fig3 is discussed in 2.2 Results section with the following Fig4 results discussion.
L,U probably, not L-U (in text)
What is Fig3M,N,O??—M and N are AFP and VFP (HeLa), but O is PFP (Fibroblasts) –why it is discussed together?
Fig3 L-M is transfected with AFP, not VFP as in the text
I don’t understand the second from the bottom paragraph on p7.
Last paragraph on p7 Fig3I:T-V is about VFP, not AFP.
2M,O,R?? does not make any sense. 3M,O,R- —the is no horizontal part in these graphs.
(Fig3P,Q)—what is about?
Fig4. The results discussion begins from 4D; 4A,B,C are missing.
In Fig4 legend in 2nd sentence: start with HeLa (A-E), continue with human fibroblasts (F), but not the other way around..
(A-E) are not transfected with VFP, but with: AFP,AFP, PFP,VFP and AFP
Only Fig4.D is described in the results
Fig4.L is explained in fig legend
Otherwise, the whole set of images in that figure is just an illustration to Fig3 second part graphs?, “movies obtained during analysis of cargo kinetics” – it is thrown into the fig4 – it has to be discussed or placed into supplemental material with the proper reference
Fig5. No name of Y axis on Fig 5N
Fig7. Graphs D-K has to be rearranged according to the text
Fig8. B,C,D,E: control images are first as photos, but the explanation begins with experimental cells, not controls. Also labeling is not specified (what is green, what is red), and
photos are placed in a random way: not red, green, red, green, but red, green, green, red. It is hard for the reader to jump back and forth.
All cells were heated for 40C?? Rewrite the comment for B-E in a logical way: control and experimental cells were heated to 40C to block the ability of COPI to form vesicles, and thus tubules, which causes the increase of AFP concentration only in IdlF cell, but not in CHO control cells
8F. is hard to comprehend; symbols are small. I believe that the black rhomb and yellow circle are misplaced????. Symbols should be reorganized in a logical way: ctrl in one group, and inhibitors are in another group.
Author Response
Our reply
Reviewer 1: The authors significantly modified the manuscript, but despite their efforts, the updated text is mostly unreadable to this reviewer. It will require a complete rewrite to be accessible to the general public. The text of the article gives an impression of the second draft version, which is halfway to the high-quality work. This version's most annoying feature is multiple figure and figure legend errors.
Reply: Thanks a lot for so deep analysis. We corrected all mistakes.
Reviewer 1: Below are some examples:
Fig. 1. L,M,O,P and are misplaced or mislabeled or both.
Reply: We corrected these mistakes.
Reviewer 1: In text: N is for L-M
Q is for O,P. meanwhile, it is written that Fig 1M,P are quantified in 1Q
1M is discussed twice: as VFP and as AFP
L-M are VFP
O-P are AFP
Reply: We corrected these mistakes.
Reviewer 1: R-X portion of Fig1 should be described in a logical way: NRK, HeLa, HF- the way it is presented as images.
1G, H are about PFP, not AFP.
The 1S symbol legends should be in logical order from highest to lowest
L-Q are all messed up in results, figure legend text and in images names:
Reply: We corrected these mistakes.
Reviewer 1: In Fig1 legend:
VFP (L,M)
AFP (O,P)
But the figure is different
In Fig1 legend:
(L,O) cryo
(O,P) nano
But the figure is different
Reply: We corrected these mistakes.
Reviewer 1: Fig.2. legend represents complete duplication of the text presented in the Results section. In fact, the is some repetition of words can be found due to the careless copy/paste for Fig. 2I
Reply: We corrected these mistakes.
Reviewer 1: Fig.3. Not clear which cells and what label was used.
Figure legend has the same text repetition issue for A-K graphs.
L-M graphs are not discussed in the Results section at all.
Fig. 3U is not explained at all.
What is (C) about?
Scale bar does not exist, even outside of the images as on Fig1. Such random placement is hard to understand.
2.2 Results. Second part of Fig3 is discussed in 2.2 Results section with the following Fig4 results discussion.
L,U probably, not L-U (in text)
What is Fig.3M,N,O??—M and N are AFP and VFP (HeLa), but O is PFP (Fibroblasts) –why it is discussed together?
Fig3 L-M is transfected with AFP, not VFP as in the text
I don’t understand the second from the bottom paragraph on p7.
Last paragraph on p7 Fig3I:T-V is about VFP, not AFP.
2M,O,R?? does not make any sense. 3M,O,R- —the is no horizontal part in these graphs.
(Fig3P,Q)—what is about?
Reviewer 1: Fig. 4. The results discussion begins from 4D; 4A,B,C are missing.
In Fig4 legend in 2nd sentence: start with HeLa (A-E), continue with human fibroblasts (F), but not the other way around..
(A-E) are not transfected with VFP, but with: AFP,AFP, PFP,VFP and AFP
Only Fig4.D is described in the results
Fig.4.L is explained in fig legend
Otherwise, the whole set of images in that figure is just an illustration to Fig3 second part graphs?, “movies obtained during analysis of cargo kinetics” – it is thrown into the fig4 – it has to be discussed or placed into supplemental material with the proper reference
Reply: We corrected these mistakes.
Reviewer 1: Fig. 5. No name of Y axis on Fig 5N
Reply: We corrected these mistakes.
Reviewer 1: Fig7. Graphs D-K has to be rearranged according to the text
Reply: We corrected these mistakes.
Reviewer 1: Fig8. B,C,D,E: control images are first as photos, but the explanation begins with experimental cells, not controls. Also labeling is not specified (what is green, what is red), and photos are placed in a random way: not red, green, red, green, but red, green, green, red. It is hard for the reader to jump back and forth.
All cells were heated for 40ËšC?? Rewrite the comment for B-E in a logical way: control and experimental cells were heated to 40C to block the ability of COPI to form vesicles, and thus tubules, which causes the increase of AFP concentration only in IdlF cell, but not in CHO control cells.
Reply: We explained all this and corrected these mistakes.
Reviewer 1: Fig. 8F. is hard to comprehend; symbols are small. I believe that the black rhomb and yellow circle are misplaced????. Symbols should be reorganized in a logical way: ctrl in one group, and inhibitors are in another group.
Reply: We remade Figures, corrected this and organized symbols according to the cargo used.
